# Prairie plants harbor distinct and beneficial root-endophytic bacterial communities

**Boahemaa Adu-Oppong**[1], **Scott A. Mangan**[2], **Claudia Stein**[3], **Christopher P. Catano**[2,4], **Jonathan A. Myers**[2], **Gautam Dantas**[1,5,6,7]*

1 The Edison Family Center for Genome Sciences and Systems Biology, Washington University in St. Louis School of Medicine, Saint Louis, Missouri, United States of America, 2 Department of Biology and Tyson Research Center, Washington University in Saint Louis, Saint Louis, Missouri, United States of America, 3 Department of Biology and Environmental Sciences, Auburn University at Montgomery, Montgomery, Alabama, United States of America, 4 Department of Plant Biology, Michigan State University, East Lansing, Michigan, United States of America, 5 Department of Pathology and Immunology, Washington University in Saint Louis School of Medicine, Saint Louis, Missouri, United States of America, 6 Department of Molecular Microbiology, Washington University in Saint Louis School of Medicine, Saint Louis, Missouri, United States of America, 7 Department of Biomedical Engineering, Washington University in Saint Louis, Saint Louis, Missouri, United States of America

* dantas@wustl.edu

**Data Availability Statement:** All sequencing files are available from NCBI BioProject PRJNA478139. All other relevant data are within the paper and its Supporting Information files.

## Abstract

Plant-soil feedback studies attempt to understand the interplay between composition of plant and soil microbial communities. A growing body of literature suggests that plant species can coexist when they interact with a subset of the soil microbial community that impacts plant performance. Most studies focus on the microbial community in the soil rhizosphere; therefore, the degree to which the bacterial community within plant roots (root-endophytic compartment) influences plant-microbe interactions remains relatively unknown. To determine if there is an interaction between conspecific vs heterospecific soil microbes and plant performance, we sequenced root-endophytic bacterial communities of five tallgrass-prairie plant species, each reciprocally grown with soil microbes from each hosts' soil rhizosphere. We found evidence of plant-soil feedbacks for some pairs of plant hosts; however, the strength and direction of feedbacks varied substantially across plant species pairs–from positive to negative feedbacks. Additionally, each plant species harbored a unique subset of root-endophytic bacteria. Conspecifics that hosted similar bacterial communities were more similar in biomass than individuals that hosted different bacterial communities, suggesting an important functional link between root-endophytic bacterial community composition and plant fitness. Our findings suggest a connection between an understudied component of the root-endophytic microbiome and plant performance, which may have important implications in understanding plant community composition and coexistence.

## Introduction

Plant-microbe interactions are increasingly implicated as key mechanisms driving plant community composition and ecosystem processes [1–7]. Plant-associated microbes can optimize

**Funding:** G.D. from the NIH Director's New Innovator Award, the National Institute of Diabetes and Digestive and Kidney Diseases (DP2DK098089) and the National Institute of General Medical Sciences (R01GM099538). B.A. was supported by National Science Foundation graduate research fellow (award number DGE-1143945). The funders had no role in study design, data collection and analysis, decision to publish or preparation of the manuscript.

**Competing interests:** The authors have declared that no competing interests exist.

nutrient uptake and pathogen exclusion, and are therefore often considered an extension of the plant genotype, phenotype, and ecological niche [8, 9]. A growing body of recent literature is focused on uncovering drivers that determine community composition of microbes within plant roots, known as the root-endophytic microbial community or root microbiome [10–21]. However, despite intense interest in the causes and consequences of plant-soil feedbacks [22–30], key gaps remain in our understanding of how root-endophytic bacterial communities influence plant-microbe interactions.

First, little is known about the ecology of root-endophytic bacterial communities in natural plant communities. Prior studies of root-endophytic bacterial communities have largely focused on agricultural or model plant species. These studies suggest that root-endophytic bacterial communities are distinct from bacterial communities found in the surrounding soil rhizosphere, and are assembled deterministically through selection by the plant host [10–17, 31, 32]. Such assembly is hypothesized to be a result of genotypic factors (e.g., innate immune system, phosphate stress response), whereby different host plants select for different microbial species, resulting in differentiation in bacterial community composition among hosts [33]. However, the degree to which similar patterns and mechanisms may contribute to the structure and function of natural plant communities remains unexplored.

Second, few studies perturb the root-endophytic bacterial community with mechanisms which drastically impact the composition. To understand the relationships between the plant and the microbiome [34–37], experiments should include treatments which affect both the microbiome and the plant. Instead, many focus on perturbations that directly affect plant community dynamics. Due to their sessile lifestyle, many plants require mechanisms to cope with abiotic and biotic fluctuations in their environment. One way for plants to cope with environmental change is through interactions with root-endophytic microbes [38, 39]. Many studies suggest that the root-endophytic bacterial community helps plants mitigate abiotic stress [32, 40–43] such as drought. This is important because the intensity and frequency of droughts are predicted to increase due to climate change [44, 45]. However, antibiotic use [46–50] is an abiotic stressor that is also increasing due to anthropogenic activities. Few studies have shown that a decrease in soil microbial diversity (as expected with antibiotic use) leads to decreases in plant performance, and even fewer have linked this to root-endophytic bacterial communities [2, 37]. Perturbations which lead to decreased variation and diversity within the soil microbial community have negative impacts on the ecosystem [51] and understanding how perturbations to the root-endophytic bacterial community impacts plant performance is crucial in creating methods to sustainably improve plant productivity [52].

Third, there are few studies which attempt to create links between the composition of the root-endophytic bacterial communities and plant-soil feedbacks; many focus on mycorrhizal fungi or treat the plant-soil microbial community as a single niche [2, 28, 29, 38, 39, 53–58]. It has been hypothesized that the plant microbiome and the plant collectively form a holobiont that influence evolution and plant community biodiversity [28, 32, 39]. This interaction is characterized as the plant-soil feedback framework due to observations in which plant species differ in their response to individual microbial species and in turn, growth rates of individual microbial species are also affected by the plant host [59]. The term feedback within the scope of this study involves 2 steps: 1) the plant host perturbs the composition of the bacterial community, and 2) this differentiation affects the performance of the plant host [6]. Plant-soil feedbacks can predict co-existence of plant species since feedbacks are plant host-specific and can either be negative or positive [22] depending on the balance of negative effects of soil-borne pathogens, herbivores, and parasites compared to positive effects of beneficial soil microbes [60]. For example, accumulating species-specific soil-borne pathogens can cause negative plant-soil feedbacks [54], thus limiting dominance and competition among plant species. In

contrast, the absence of species-specific soil-borne pathogens, for example in disturbed environments [23, 61–63], can allow plant species to increase in abundance and accelerate competitive exclusion [64]. However, few studies have intensely and carefully examined how root endophytic bacterial communities can either partially or wholly explain plant-soil feedbacks [32].

In this study, we focused on plant species which commonly co-exist within the native North American prairie ecosystem to understand the ecology of their root endophytic bacterial communities and determine the extent to which the root endophytic bacterial community composition contributes to their co-existence. The North American prairie ecosystem is one of the Earth's most endangered ecosystem [65]. We sequenced root-endophytic bacterial communities of five tallgrass-prairie plant species, each reciprocally grown with soil microbes from each hosts' soil rhizosphere. We addressed four questions: 1) Does the composition of root-endophytic microbial communities differ among host species of co-occurring prairie plants?; 2) Are differences in the performance of conspecific plants associated with differences in root-endophytic bacterial community composition, and if so, do certain bacterial species drive plant performance?; 3) To what extent do perturbations to the soil microbial community disrupt associations between root-endophytic bacterial communities and the host?; and 4) Is there evidence of plant-soil feedback within these tallgrass-prairie species, and if so, is the root endophytic bacterial community driving the feedback? We conducted a plant-soil feedback study with soils collected from plant species commonly found in the tallgrass-prairie ecosystem. To address these questions, we sequenced the 16S rRNA gene from the endophytic root compartment of plants which were initially grown in sterile conditions and compared the bacterial community composition to the data collected from the plant-soil feedback study.

## Materials and methods

### Plant species and soil collection

We chose 5 prairie species, 4 natives: *Monarda fistulosa* (Wild Bergamot), *Ratibida pinnata* (Grey-head coneflower), *Heliopsis helianthoides* (Smooth oxeye), *Conyza canadensis* (Horseweed); 1 invasive which was listed as a noxious weed in Missouri (USDA 2019): *Carduus nutans* (Musk Thistle). These plant species were chosen because they are highly abundant in prairies in the Midwestern USA; therefore, we could collect enough plant-associated field soil to conduct the greenhouse experiment. We purchased all seeds from Prairie Moon Nursery (Winona, Minnesota, USA) with the exception of *Carduus nutans* seeds, which were collected at Tyson Research Center (Eureka, MO, USA) in June 2013.

Experimental prairie restoration plots were established within a 0.5 ha field at Washington University's Tyson Research Center, Missouri, USA, located south-west of St. Louis. The climate of this area is warm and temperate, with 897mm annual precipitation and 13.7˚C annual temperature. Soils are limestone derived and clay rich. The history of our field site represents a good match for sites targeted for prairie restoration. Prior to 1984, the study area was used as an agricultural or hay field. From 1984 to 1989, the study area was an experimental corn field. Throughout the course of the lifecycle of the experimental plots (2009–2016), the field was managed with practices generally used in prairie restoration. The entire field was mowed in June & August 2009. Late winter or early spring burns were performed in 2011, 2013 and 2016. Three non-native species that display invasive behavior (*Carduus nutans*, *Vicia villosa* and *Sorghum halepense*) were removed manually or with a targeted herbicide (40% glyphosate). The plots were seeded with 25 Missouri ecotype native forb and 5 grass species, the seeding densities and common names can be found here along with more information about the experimental prairie restoration plots [66].

Step 1 of the plant-feedback framework is to create differentiated soil communities by either allowing plant hosts to grow in similar initial soil communities for a few months or sampling close to adult plants in field sites due to the short generation time and rapid community dynamics of microbial communities [54]. Step 2 is to measure the performance of the plant host, by growing the plants in an inoculation of the differentiated soil communities surrounded by a common background soil to isolate microbial effects [54]. We chose to collect soil from these experimental plots to serve as our differentiated soils, which we will refer to as soil history throughout the manuscript. By measuring plant performance to differentiated soil communities, we can estimate net soil community feedback parameters. During the summer of 2013, soils used as our differentiated soils (see below) were obtained from the rooting zone (< 1m from base of stem) of patches of mature individual plants for each species from 5 different prairie restoration experimental plots located at Tyson Research Center and soil samples were not pooled across the experimental plots. Shovels used to collect soil were cleaned and sterilized with 70% (v/v) ethanol and a blow torch in between plant host inoculum collection to avoid cross-contamination. We chose experimental prairie sub-plots that were not manipulated with chemicals (no phosphorous or fertilizer added). Bulk soil was collected 30m away from the experimental Tyson plots to serves as our common background soil. All soils were stored in the dark at 4˚C.

## Overview of greenhouse experiment

We conducted a full reciprocal greenhouse experiment where we exposed sterile seedlings of each plant host to either their own soil microbial community (conspecific) or to microbial communities associated with each of the other plant hosts (heterospecific) (Fig 1). This reciprocal design allowed us to investigate the extent in which microbial community assembly was a function of deterministic host selection or random assortment. We controlled for abiotic soil effects to better link growth responses to the differentiated soil microbial communities by filling all pots with the same background soil (2:1 bulk soil-sand mix) that was autoclaved twice (gravity cycle for 65 min) [67]. We then added a small quantity (6% of pot volume– 6” diameter pots) of field-collected conspecific or heterospecific inoculum (soil history) for each plant host. To ensure that roots were colonized by microbes in the collected inoculum, we surface sterilized and germinated seeds in autoclaved (gravity cycle for 65 min twice) Propagation Mix (Sungro horticulture Agawam, MA, USA). Germinated seedlings were transferred to the individual pots using sterilized tweezers and scoopula. Fourteen replicates of each plant host received heterospecific inoculum. Twenty-four replicates of each plant host received conspecific inoculum. Six replicates for each plant host received conspecific and heterospecific autoclaved inoculum. Half of all replicates were subjected to an antibiotic treatment which allowed us to further test the strength of deterministic factors on root endophytic bacterial community composition. This resulted in 5 (plant hosts) x [4 (heterospecific inoculum) x 2 (antibiotic treatment) x 7 replicates + [1 (conspecific inoculum) x 2 (antibiotic treatment) x 12 replicates]] + [5 (plant hosts) x 5 (autoclaved inoculum) x 2 (antibiotic treatment) x 3 replicates] = 550 experimental units in a semi-full factorial design (Fig 1).

## Perturbations: Autoclaving and antibiotics

Autoclaving soil perturbs the microbial community by reducing the number of bacterial species in a community (S1A Fig, S1A Table). We autoclaved half of the collected inoculum (gravity cycle for 65 min followed by a second gravity cycle for 65 min 24 hours later).

Antibiotics were chosen as a perturbation due to their ability to directly affect microbial communities by eliminating species from the communities without directly impacting plant

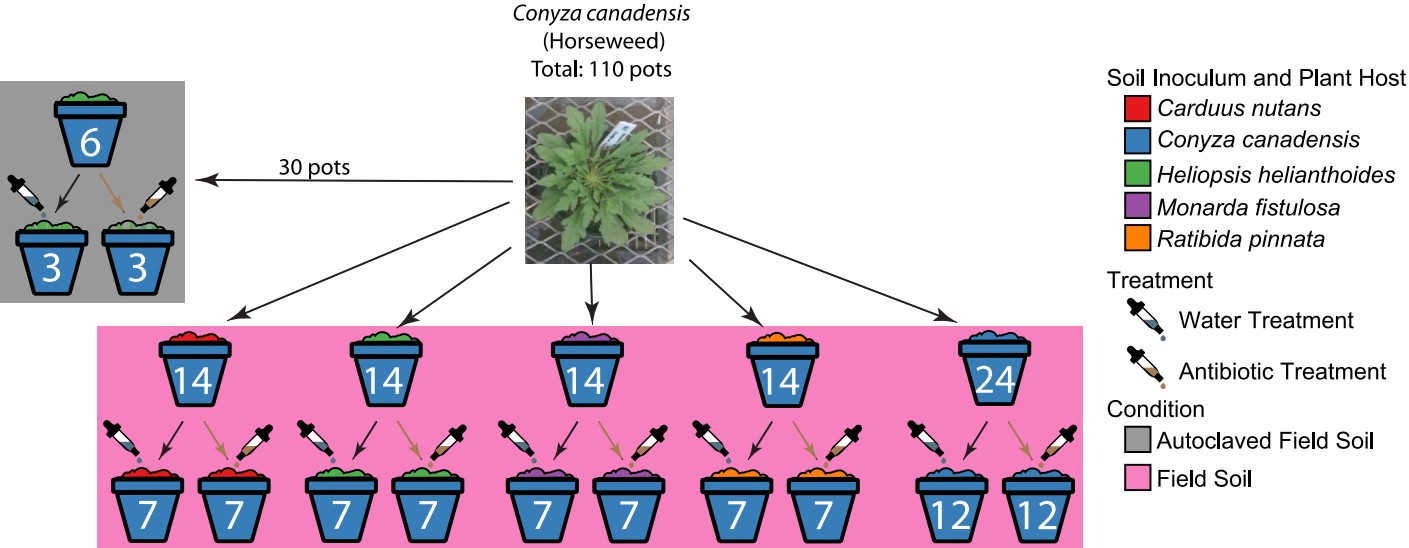

**Fig 1. Illustrative description of study design.** Only one species is represented in the picture; however, all plant hosts underwent the same manipulations. Seeds from each plant host (color of which plant host is depicted by color of the pot) were grown in soil collected from all plant hosts (color of source inoculum is depicted by the color of the soil). Soils were subjected to both an antibiotic treatment (depicted by color of liquid in the pipette) and autoclaved treatment (depicted by striations of the soil).

growth (S1 Fig). Plant performance was not affected when grown in the presence or absence of antibiotics (S1B Fig). We chose four antibiotics: chloramphenicol (8mg/L), oxolinic acid (0.2 μg/mL), gentamicin (32mg/L or 4mg/L), streptomycin (512mg/L). Chloramphenicol and gentamicin are used in agar plates when isolating fungi to decrease the presence of bacteria [68]. Oxolinic acid, gentamicin, and streptomycin are used in the plant-agriculture community to target bacterial pathogens that affect crops [69]. Chloramphenicol is a broad range antibiotic that is bacteriostatic and inhibits protein synthesis by binding to the 50S ribosomal subunit (Sigma Product Information). Oxolinic acid is effective against gram-negatives and is a quinolone compound. It inhibits the DNA gyrases (Sigma Product Information). Gentamicin is a broad range antibiotic that inhibits bacterial protein synthesis by binding to the 30S subunit of the ribosome (Sigma Product Information). Streptomycin is a broad range antibiotic but has been known to be less effective against Gram-negative aerobes [70]. It blocks protein synthesis by targeting the 70S ribosome. The concentration of the antibiotics were determined by using the highest MIC concentration from EuCast2 (https://mic.eucast.org/Eucast2/SearchController/). Pots not treated with antibiotics were administered 10ml of autoclaved deionized water. The first treatment was given July 12, 2013; we administered 10ml of the antibiotic cocktail. For the rest of the treatments we administered 15ml of the antibiotic cocktail every 2 weeks.

## Plant care and trait measurement

The experiment started July 2013 and ended October 2013. The duration was chosen to ensure all plants had enough time within the vegetative stage. Only one plant, *H. helianthoides*, flowered during this time period. All pots were arranged twice into randomized blocks and maintained in the greenhouse for the duration of the experiment. Dropped leaves were collected and included in total biomass for the individual. At the end of the experiment, we harvested both shoot and root separately. Roots were carefully washed with water over a 500-μm sieve to

remove all soil particles. Shoots and roots were placed in separate envelopes. We measured dried biomass after oven-drying the samples at 60°C for 48 hours.

## Calculating plant-soil feedback interaction

To calculate plant-soil feedback interaction coefficient ($I_s$), we used the dried biomass of both above-below ground plant parts and calculated the interaction using this equation [71]:

$$I_s = \alpha_A - \alpha_B - \beta_A + \beta_B$$

$\alpha_A$ is the effect that plant species A has on itself while $\alpha_B$ is the effect that plant species A has on B. $\beta_B$ is the effect that plant species B has on itself while $\beta_A$ is the effect that plant species B has on A. We demonstrate how we calculated the interaction coefficient using an example with *H. helianthoides* and *C. nutans*. Total dried biomass of *H. helianthoides* grown in inoculum collected from conspecifics = $\alpha_A$. Total dried biomass of *H. helianthoides* grown in inoculum collected from *C. nutans* = $\beta_A$. Total dried biomass of *C. nutans* grown in inoculum from *H. helainthodies* = $\alpha_B$. Our final value variable, $\beta_B$, is the total dried biomass of *C.nutans* grown in conspecific inoculum. We did this calculation for each of the pairs and the average of the pairs for a plant host served as our main plant-soil feedback interaction coefficient. We used a one-sample t-test to determine if the feedback interaction coefficient was significantly different from 0.

## Characterization of root endophytic bacterial communities

To characterize the root endophytic bacterial communities, we weighed approximately one gram of belowground biomass for microbial extraction and stored it at -80°C. Sterility was maintained between samples by placing roots in sterilized weigh boats and only one plant individual was measured at a time to limit cross-contamination. The selection of the root sample was standardized to the secondary root. Since we wanted to limit the amount of cross contamination, we did not standardize the exact location of the extracted root sample across all plants. To accurately measure belowground biomass, total belowground biomass was weighed before and after removal of the portion used for microbial extraction. The estimated loss was calculated and added to the dried biomass weight.

   Belowground biomass was resuspended in 15ml of filter sterilized PBS-S buffer (130mM NaCl, 7mM Na2HPO4, 3mM NaH2PO4, pH 7.0, .02% Silwet L-77) and sonicated (Fisher Scientific Sonic Dismembrator Model 500, Pittsburgh, PA, USA) at low frequency for 5 min with five 30 sec bursts followed by five 30 sec rests for 252 root samples. We collected 14 samples (After Sonication) after this stage and submitted them for sequencing. Then roots were resuspended in 15 ml of filter sterilized PBS-S buffer and centrifuged at 1,500g for 20 minutes. We collected another 14 samples (After Wash) after this stage and submitted them for sequencing. We sonicated and washed the roots to remove any bacteria that could be found on the external surface of the roots. The roots were aseptically transferred to a new 15ml and freeze dried overnight. We validated that the sonication and washes were sufficient in removing surface and exterior bacteria by sequencing these extracts and determining that the microbial communities were significantly different (S2A Fig). The microbial community was extracted from roots per manufacture's protocol using the PowerSoil DNA Isolation Kit (Mo-Bio Laboratories, Carlsbad, CA, USA). We performed PCRs in triplicates to control for bias in PCR reactions and amplified the 16s rRNA V4 region (http://www.earthmicrobiome.org/emp-standard-protocols/16s/) using the barcodes designed in [72]. Before sequencing, we visualized the bands on gels. After a positive confirmation, we combined all samples and sequenced them on the Illumina MiSeq platform (Illumina Inc., San Diego, CA, USA) with 2x250 bp paired-end

reads at the Edison Family Center for Genome Sciences at Washington University in St. Louis. Sequences were demultiplexed using QIIME [73]. Paired-end reads were truncated at the first base with a quality score of <Q4 and then merged with usearch [74], with a 100% identity in overlap region and a combined length of 253±5 bp. The merged reads were then quality filtered by usearch with a maximum expected error of 0.5. Operational taxonomic units (OTUs) were picked using the usearch pipeline [74] and known chimera OTUs were filtered from the list. Reads were matched to OTUs at 97% sequence identity. Representative sequences from each OTU were aligned using PyNAST and assigned taxonomy using RDP Classifier using QIIME version 1.5.0-dev. OTUs which matched chloroplast or mitochondria were removed from the dataset. Any sample with fewer than 30 OTUs were dropped from the study. Additionally, OTUs which were found in only one sample or had fewer than 30 individuals were removed from the dataset for a total of 595 OTUs. The data from the Illumina sequencing will be deposited in NCBI BioProject PRJNA478139 upon journal acceptance.

Microbial community count data were transformed using the DESeq2 package in R based on previous recommendations [75, 76]. All analyses were performed using the package 'vegan' v.2.4.1 [77], 'RVAideMemoire' v.0.9.61 [78] and 'phyloseq' v.1.18.1 [79] in R version 3.2.2. Principal coordinates (PCoA) of Bray-Curtis pairwise dissimilarities were identified using the vegan function 'capscale'. To explain the difference in dissimilarity of microbial communities, we tested the effect of host, soil history, autoclaving of field soil and exposure to antibiotics in a full model using the non-parametric permutation test ADONIS II in package 'RVAideMemoire' with 999 permutations. The $r^2$ value from the ADONIS reflects the amount of variation in microbial community composition explained by each of the factors tested. We corrected for multiple comparisons with the False Discovery Rate post-hoc test to determine which pairs were significantly different.

The aovp function in the lmPerm package which performs a fitting and testing ANOVA using permutation test was used to determine differences between alpha diversity of the root endophytic bacterial communities and total dried biomass between the plant hosts.

To find taxa that were differentially abundant in our samples according to our classifications, we used the package 'ANCOM' v.1.1.3 [80]. 'ANCOM' first compares the log-ratio of the abundance of each taxon to all remaining taxon one at a time and then Mann-Whitney U is calculated on each log ratio [81]. This method when compared to other differential abundance based statistical tests does not have inflated average FDR [81]. We then conducted a pairwise-t-test with Bonferroni adjusted p-values to determine which pairs where significantly different.

## Linking belowground species composition to plant performance

Measuring plant-soil feedbacks only involve measuring biomass and manipulating the soil inoculum (soil history). We additionally correlated the composition of the bacterial communities to plant performance to understand if differences in endophytic root bacterial compositions could explain differences in plant biomass. We log-transformed biomass to satisfy the linear model assumption prior to assessing treatment effects on biomass for different plant hosts [67]. We conducted an ANOVA to test for the effect of autoclaving of field soil, exposure to antibiotics, plant host and soil history. We also tested for the effect of interactions between plant host and sub-plot location of soil collection to ensure soils collected in different sub-plots did not affect biomass. To link performance to composition of root endophytic bacterial communities in perturbed states, we tested for an interaction between plant host and autoclaving of field soil and plant host and soil history. First, we determined which OTUs were differentially abundant in the various source history by using the package 'ANCOM' v.1.1.3 [80]. To

test whether differences in performance could result in turnover in root endophytic bacterial communities, we correlated composition of root endophytic bacterial communities and biomass. We used a Mantel test with 999 permutations in the package 'ade4' v.1.7–4 to test the significance of the correlations. To test whether a taxon of bacteria could affect biomass, we correlated biomass and abundance of bacteria taxon. We used cor.test with pearson correlations in the package 'stats' v.3.3.2. P values were adjusted using Bonferroni. All results were graphed using 'ggplot2' v.2.2.0 [82] in R version 3.3.2.

## Results

### Ecology of root-endophytic microbial communities in natural plant communities

**Host identity structures root endophytic bacterial communities more than soil history.** To investigate if variation between root endophytic bacterial communities is explained by the plant host, Bray-Curtis was calculated on each sample and ADONIS was used to determine the amount of variation explained by plant host and soil origin. We found that root endophytic communities are plant host-specific, independent of whether they are grown in soil microbial communities associated with similar or different hosts. In a subset of samples which were non-autoclaved and non-antibiotic treated, the compositional differences in root bacterial community was better explained (as indicated by the higher $r^2$ value) by plant identity (ADONIS $p < 0.001$, $r^2 = 0.11$, Fig 2A) than by soil history (ADONIS $p < 0.001$, $r^2 = 0.03$, Fig 2B). When accounting for all samples which were autoclaved (ADONIS $r^2 = 0.07$) and treated with antibiotics (ADONIS $r^2 = 0.005$), more of the variation was explained by plant host identity (ADONIS $p < 0.001$, $r^2 = 0.07$. Fig 2C) than by soil history (ADONIS $p < 0.001$, $r^2 = 0.02$ Fig 2D, S2A Table). To determine if there were changes in abundance for specific bacterial species between the plant hosts, we used ANCOM to detect differentially abundant taxa. Of the approximately 100 OTUs which were differentially abundant between plant hosts (S3 Fig), 67% belonged to the Proteobacteria phylum, and 13% were exclusively found in one plant host (S3 Table).

**Soil history influences plant performance and root endophytic bacterial communities.** To determine if specific soil histories affected plant performance, we measured the dried biomass of all plants grown in each of the field-soil inocula. We found that soil history, autoclave treatment, and plant host had significant effects on plant biomass (S1B Table). Tukey's *post hoc* tests indicated that total plant biomass was greater in the soil that was collected underneath *M. fistulosa* when compared to plants grown in soil collected underneath other plants (Fig 3A, S4 Table). To determine if the variation between the microbial communities could be explained by soil history, we measured *β* diversity using Bray-Curtis dissimilarities. Permutational multivariate analysis of variance using distance matrices (ADONIS) confirmed that a portion of the variation was explained by soil history (Fig 2B, S2A Table). Soil history explained only 1.9% of the variation in root endophytic communities (S2A Table) which suggests that the soil history is significant but has a weak effect on the assembly of the endophytic microbial community.

To investigate if the overall differences in the endophytic bacterial communities could be traced to specific microbes, we used analysis of composition of microbiomes (ANCOM) to detect differentially abundant taxa. We identified three bacterial species that are identified using this method and we were able to classify the species to the family level: Planococcaceae (OTU 1321), Cytophagaceae (OTU 17), and Micromonosporaceae (OTU 87) (Fig 3B, S5 Table). Planococcaeceae was depleted in *C. canadensis'* soil history and Micromonosporaceaea was depleted in *C. nutans'* soil history. Cytophagaceae was enriched in *C. canadensis'* and *H.*

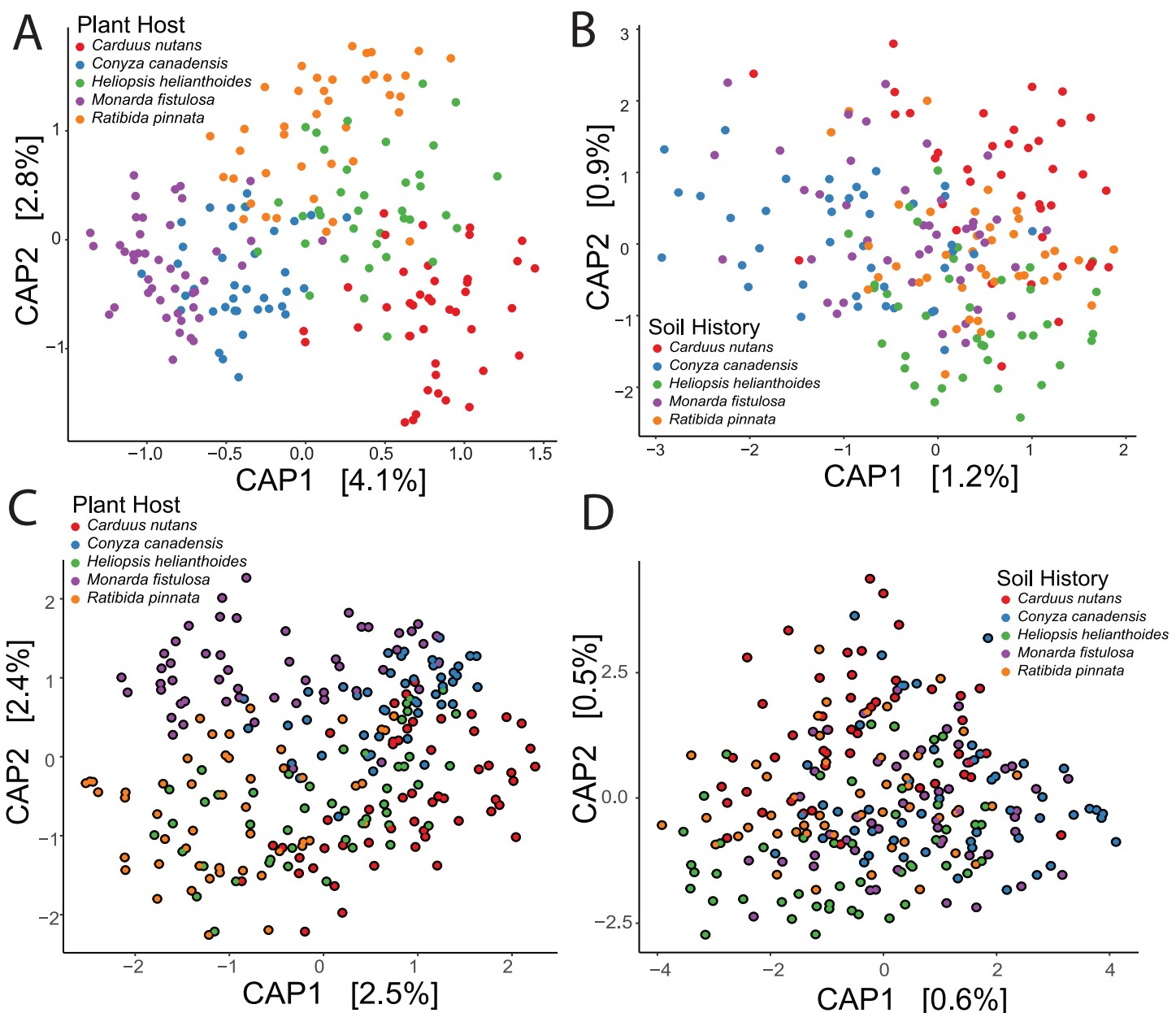

**Fig 2. Plant host explains variation in root endophytic bacterial community.** (A) CAP plot showing the live soil endophytic root microbiome of *Carduus nutans* (red), *Conyza canadensis* (blue), *Heliopsis helianthoides* (green), *Monarda fistulosa* (purple) and *Ratibida pinnata* (orange) [ADONIS p < 0.001, $r^2$ = 0.11, n = 201]. (B) CAP plot showing the endophytic root microbiome clustered by soil history [ADONIS p < 0.001, $r^2$ = 0.03, n = 201] (C) CAP plot showing both the live and autoclaved soil endophytic root microbiome clustered by plant host [ADONIS p < 0.001, $r^2$ = 0.07, n = 247] (D) CAP plot showing both live and autoclaved soil endophytic root microbiome separated by soil history [ADONIS p < 0.001, $r^2$ = 0.02, n = 247].

*helianthoides'* soil history. Our results indicate a weak effect of soil history on plant host and the composition of the endophytic microbial community.

**Differences in plant performance correlate with differences in microbial community composition.** To understand the effect of root-endophytic bacterial community composition and diversity on plant performance for individual plant species, we measured the dried weight of plants grown in autoclaved and non-autoclaved soils. Overall, plant biomass was reduced in autoclaved soil (*Anova* p < 0.0001, S1B Fig, S1B Table). We confirmed that changes in plant

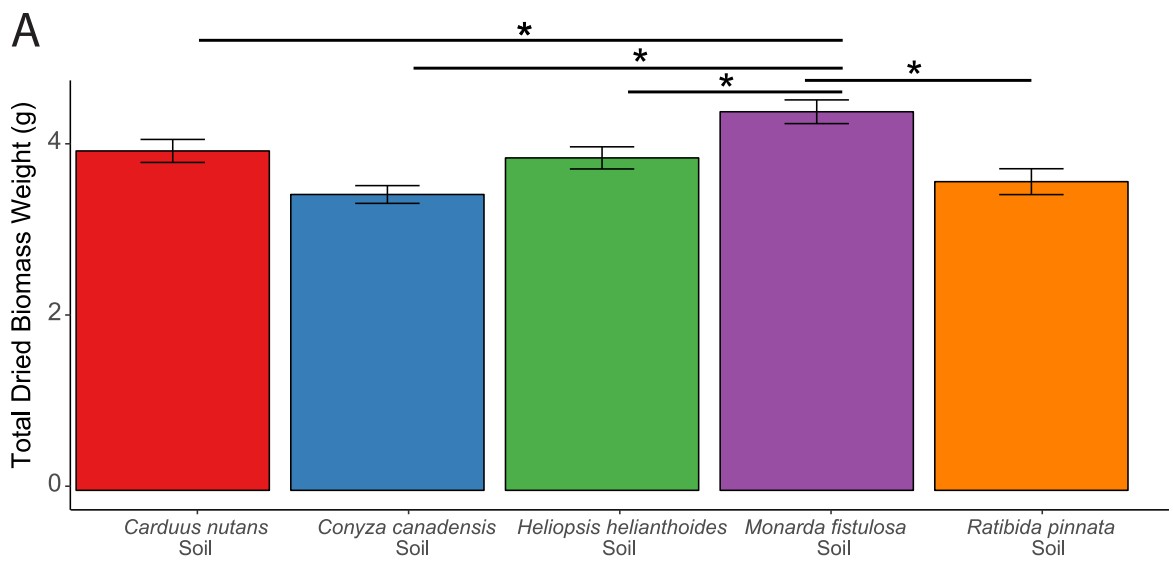

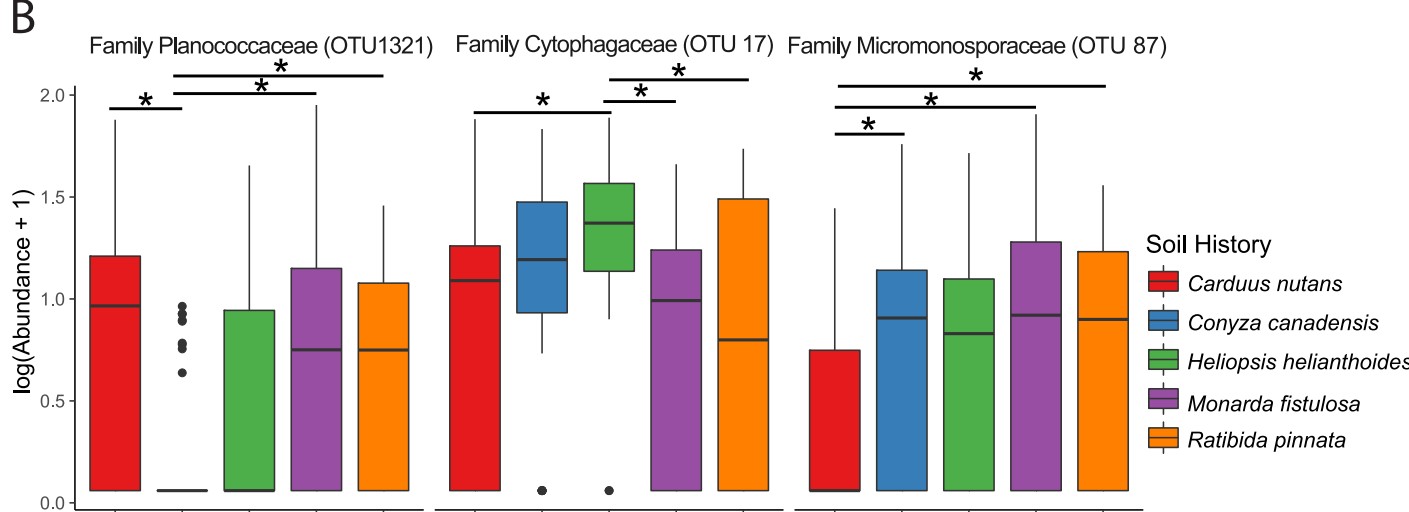

**Fig 3. Plant biomass is affected by soil treatment.** (A) Bars represent mean total dried biomass across all field soils and all autoclaved field soils with SE. The asterisks indicate statistically significant differences [ANOVA tests with $P \leq 0.05$, n = 537]. (B) (C) Box plots of differentially abundant OTUs found across all live soil history [ANCOM (Mann-Whiteny U + FDR correction $P \leq 0.05$, n = 201)].

biomass can be attributed to the soil biotic rather than abiotic components because biomass when summed across all plant species in each autoclaved soil was uniform (S2B Fig). We can attribute plant performance to the soil biotic component because we controlled for potential variation in abiotic properties introduced by the small volume of soil history. Additionally, we assessed seedling growth in the same plant-inoculum combinations but with autoclaved inoculum. Furthermore, plant performance responded to the autoclave treatment in a plant species-specific manner. Three plant hosts had lower biomass in autoclaved field soils: *M. fistulosa*, *H. helianthodies*, and *R. pinnata* (Fig 4A, S6 Table). In contrast, *C. nutans* and *C. canadensis* had equivalent fitness in field soils and autoclaved field soils (Fig 4A) suggesting that our invasive, *C. nutans*, does not demonstrate strong plant biomass relationships with the microbial communities (neither beneficial nor inhibitory) in the prairie system. As for *M. fistulosa*, *R. pinnata*, and *H. helianthoides*, which are all native (non-weedy) species, the reduction in biomass

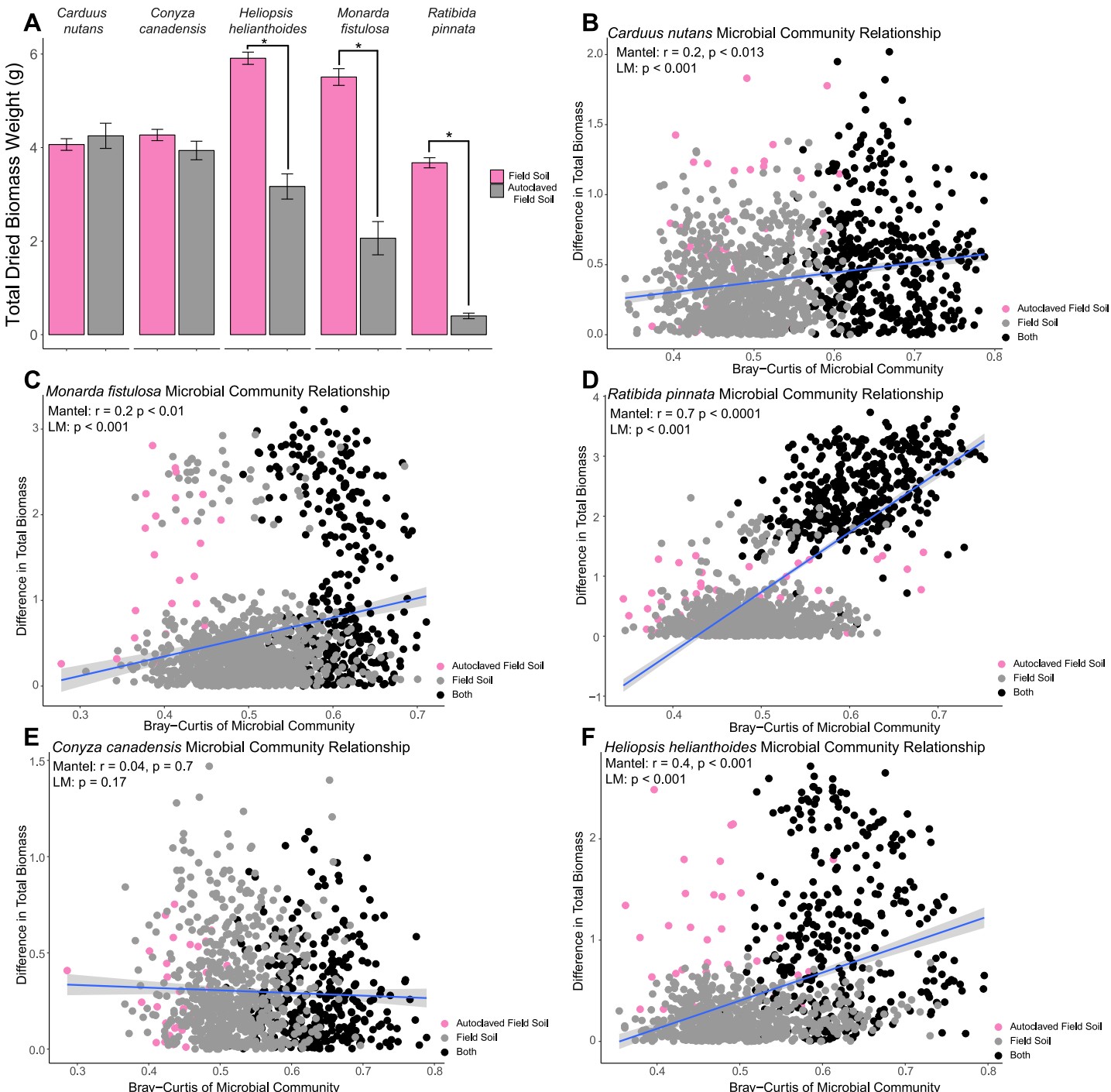

**Fig 4. Root-endophytic bacterial community composition as a function of growth differences.** (A) Bars represent mean total dried biomass across all field soils and all autoclaved field soils with SE. The asterisks indicate statistically significant differences [ANOVA tests with $P \leq 0.05$, n = 537]. Difference in total biomass between individuals of the same plant species correlated with difference in endophytic root microbiome composition (calculated using Bray-Curtis dissimilarity). Each point represents two individuals' difference in root microbiome and biomass. (B) *C. nutans* (n = 52), (C) *M. fistulosa* (n = 52), (D) *R. pinnata* (n = 50), (E) *C. canadensis* (n = 46), (F) *H. helianthoides* (n = 47). Differences calculated between samples grown in autoclaved soil are in grey, between samples grown in field soil are in pink, and between one sample grown in field soil and one grown in autoclaved soil in black. Lines and regression statistics are based on Mantel and linear regression.

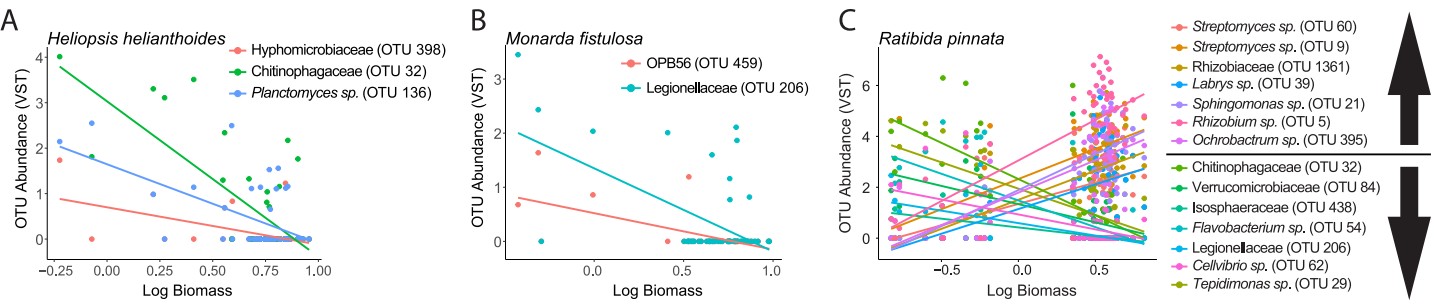

**Fig 5. Bacterial species abundance with statistically significant (P ≤ 0.05, after Bonferroni correction) correlation with plant log biomass.** Bacterial species within the roots of (A) *H. helianthoides* (B) *M. fistulosa* and (C) *R. pinnata*.

when grown in autoclaved field soils suggests the potential beneficial relationship between the plants (biomass) and soil microbial communities. We estimated this potential plant root-microbiome effect by correlating plant biomass to the divergence of host-specific root endo-phytic bacterial communities. We discovered that conspecifics with similar biomass have even more similar bacterial community compositions. *C. nutans* (p < 0.013), *H. helianthodies* (p < 0.001), *M. fistulosa* (p < 0.01,), and *R. pinnata* (p < 0.0001), but not *C. canadensis* (p = 0.7), demonstrated a correlation between biomass and community similarity (Fig 4). Taken together, these results suggest that plant biomass can be affected by the composition of the endophytic bacterial community. However, further work will be needed which exclusively manipulate endophytic root communities to determine the effect of the endophytic bacterial community on plant fitness.

To determine if specific bacterial taxa were correlated with, and hence potentially responsible for, measured differences in plant biomass, we regressed the abundance of bacterial species against the biomass of each plant host. We observed significant correlations for specific taxa in *H. helianthoides* (Fig 5A), *M. fistulosa* (Fig 5B), and *R. pinnata* (Fig 5C) roots. Interestingly, each of the five plant species tested responded differently to the abundance of different bacterial species. Therefore, if individuals of the same plant species are affected by the same species-specific pathogens (e.g. root endophytic bacterial taxa), then that could lead to negative feed-backs and restrict proliferation of conspecifics [43]. To directly test this hypothesis for root endophytic bacteria, future studies should focus on characterizing root endophytic bacterial communities in the presence or absence of competition with plant conspecifics and heterospecifics.

## Perturbations to the soil microbial community did not disrupt associations between root-endophytic bacterial communities and the host

To understand the specificity of the root endophytic microbial community to the host, we measured the change in root endophytic microbial community composition after exposure to antibiotics (ADONIS p = 0.26, $r^2$ = 0.005) and the autoclave treatment (ADONIS p < 0.001, $r^2$ = 0.07). Both influenced the overall community composition of the bacterial community (S2A Table) but only the autoclave treatment affected plant biomass for 3 of the 5 species (see section above). Interestingly, the root endophytic microbiota of plants grown in autoclaved soil clustered by plant host (ADONIS p < 0.001, $r^2$ = 0.24, S2C Fig, S2B Table), suggesting that even under extreme perturbation, endophytic microbial communities are structured primarily by the plant host.

## Pairwise plant-soil feedback did not correlate with root-endophytic bacterial communities

To determine the strength of a plant-soil feedback, we calculated plant performance when grown in conspecific vs heterospecific soils (S4 Fig). Plant performance decreased when grown in heterospecific soils for *M. fistulosa* while it increased for *C. canadensis* (S7B and S7C Table). We used this information to calculate feedback which is plant performance of both target plant hosts and heterospecific plant hosts grown with soil biota collected from conspecifics vs from heterospecifics. The resulting interaction between host and soil history was used to define strength and direction of feedback (S5A–S5E Fig). We did not measure any plant-soil feedback within our overall study system (S5F Fig). We did measure significant plant-soil feedback for 4 pairs and 2 were treading towards significance (Fig 6). However, all but one (Fig 6J) did not show any correlation between root-endophytic microbial communities and plant biomass suggesting that root-endophytic microbial communities do not mediate plant-soil feedbacks.

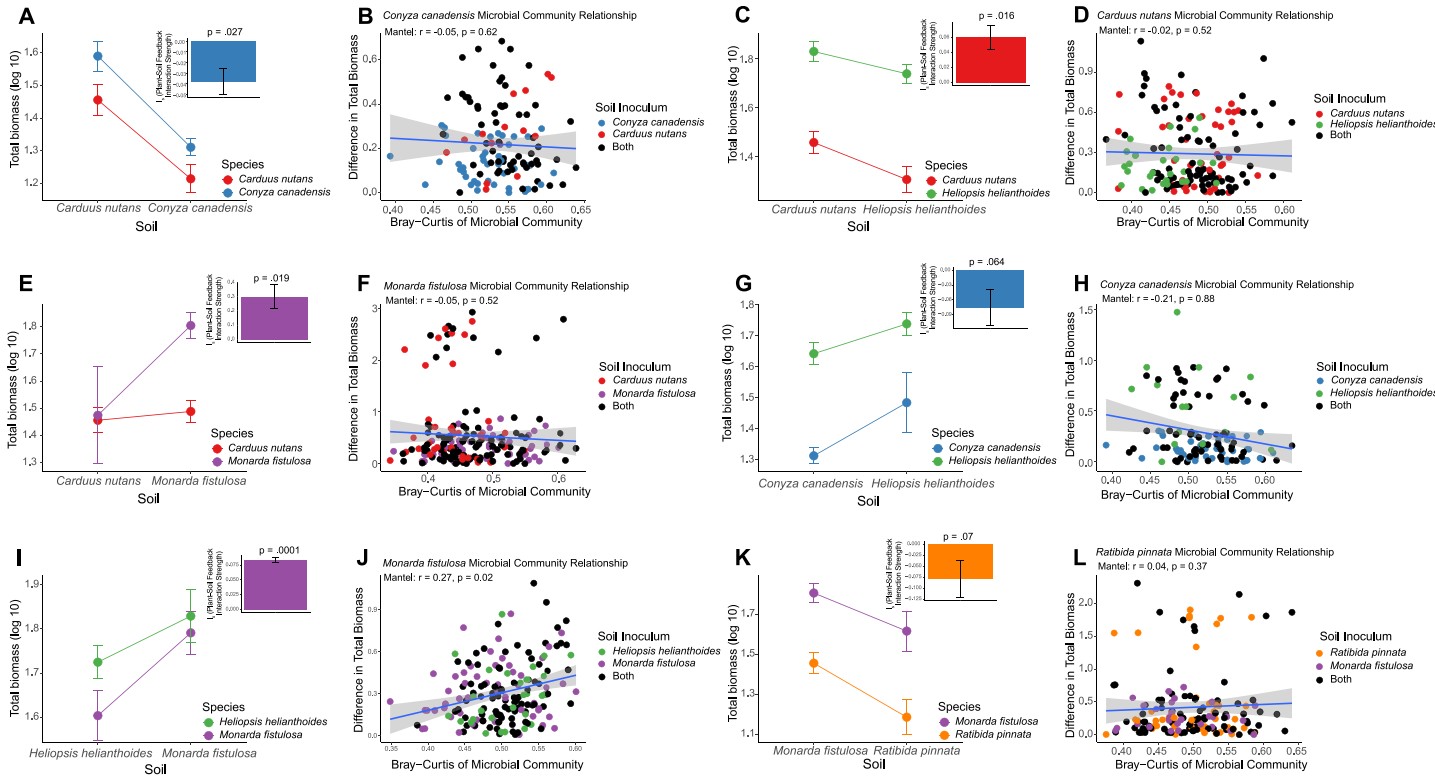

**Fig 6. Significant plant-soil feedback pairs correlated with root-endophytic bacterial community composition.** Plant performance when grown in soil history from conspecifics vs heterospecifics: (A) Performance of *C. nutans* (n = 37) and *C. canadensis* (n = 38). The resulting interaction between plant host and soil history defined a significant negative feedback (insert, p = 0.027). (C) Performance of *C. nutans* (n = 37) and *H. helianthoides* (n = 38). The resulting interaction defined a significant positive feedback (insert, p = 0.016). (E) Performance of *C. nutans* (n = 37) and *M. fistulosa* (n = 38). The resulting interaction defined a significant positive feedback (insert, p = 0.019). (G) Performance of *C. canadensis* (n = 38) and *H. helianthoides* (n = 38). The resulting interaction defined a closely significant negative feedback (insert, p = 0.064). (I) Performance of *M. fistulosa* (n = 37) and *H. helianthoides* (n = 38). The resulting interaction defined a closely significant positive feedback (insert, p = 0.0001). (K) Performance of *M. fistulosa* (n = 38) and *R. pinnata* (n = 37). The resulting interaction defined a closely significant negative feedback (insert, p = 0.07). Difference in total biomass grown in conspecific vs heterospecific soil correlated with difference in endophytic root microbiome composition (calculated using Bray-Curtis dissimilarity). Each point represents two individuals' difference in root microbiome and biomass: (B) *C. canadensis* grown in inoculum from *C.nutans* vs conspecific soil (D) *C. nutans* grown in inoculum from *H. helianthoides* vs conspecific soil (F) *M. fistulosa* grown in inoculum from *C. nutans* vs conspecific soil (H) *C. canadensis* grown in inoculum from *H. helianthoides* vs conspecific soil (J) *M. fistulosa* grown in inoculum from *H. helianthoides* vs conspecific soil (L) *R. pinnata* grown in inoculum from *M. fistulosa* vs conspecific soil.

## Discussion

### Soil history and plant host affect the community structure of root-endophytic bacterial community

While selection by the plant host have been proposed as mechanisms structuring the soil microbial community [33, 43, 83], we provide evidence that the root-endophytic bacterial community is largely structured by the plant host, regardless of variation in soil history. Our results are consistent with several studies that have made this link between plant species identity and soil bacterial community [84–86] as well as genotype and soil bacterial community [12, 17, 87, 88]. Since manipulating the genome of model plants such as *Arabidopsis* is achievable, other studies have documented phylum-level differences in the endophytic bacterial community structure of *Arabidopsis* mutants [89], showing that under an artificial system the composition of the root endophytic bacterial community is altered by the genotype of the plant host. Within our study, we discovered that individuals within each plant host are more similar to each other than to different plant hosts (Fig 2A). Moreover, since we used natural variation within each plant host rather than artificially manipulating the plant genome and noticed large variation between individuals within each plant host indicating that root endophytic bacterial microbiome is not identical within all individuals of the same plant host (Fig 2A).

There were only three bacterial taxa which were differentially abundant in non-autoclaved field soils and these taxa were not exclusively enriched in one soil source (Fig 3B). Interestingly, a Bacteriodetes family, Cytophagaceae, was differentially abundant across plant hosts and soil source; however, we only found one OTU that exhibited this characteristic. This corroborates theories that microbial taxa create non-random distributions [90] which is consistent with studies showing that root endophytic bacterial communities are very similar regardless of soil source [10, 11, 13].

Many studies have highlighted the enrichment of Actinobacteria within the root endophytic microbial community [10, 11, 43, 91]; however, we note the importance of Proteobacteria in distinguishing root endophytic microbial communities [92] (S3 and S6 Figs). We observed that the dominating phyla across all root endophytic bacterial communities in this study in decreasing abundance are Proteobacteria, Firmicutes, Bacteriodetes, and Actinobacteria (S6 Fig), which have all been reported as dominant members of various root endophytic bacterial communities [33]. In contrast to our prairie plant root microbiomes, the dominant phyla of the root microbiome of the model plant *Arabidopsis thaliana* in decreasing abundance are Actinobacteria, Proteobacteria, Bacteriodetes, and Firmicutes [11], which likely reflects ecological and environmental differences between these model and non-model plant species. We observed that about 10% of the Proteobacteria that were differentially abundant were exclusively found in one plant host. *Burkholderia bryophila* (OTU 45) which is a known anti-fungal against phytopathogens and a plant-growth-promoter was exclusively enriched in *C. nutans*, an invasive of prairie communities. Further studies using *B. bryophila* as an inoculum might help elucidate whether this or other bacterial species contribute to plant performance either negatively or positively.

### Root-endophytic bacterial communities could potentially affect plant performance

Many studies have tightly linked increases in diversity of soil microbial communities to plant performance [93–95], while within this study we link performance to certain microbial species as well as the root-endophytic microbial community rather than treating the soil community

as one black box. We observed a correlation between differences in plant biomass with differences in the composition of the root-endophytic bacterial community composition except *C. canadensis* (Fig 4B–4F). Two of the taxa that were enriched in high biomass samples (Fig 5), *Ochrobactrum* sp. and *Sphingomonas* sp., have been identified as potential growth enhancing bacteria in previous experiments [96, 97]. Additionally, the depletion of certain OTUs belonging to the families *Planctomycetaceae*, *Legionellaceae*, and *Chitinophagaceae* in low biomass samples was consistent across plant species. These bacteria may be candidates as potential plant-specific growth inhibitors. However, the evidence presented in this study is based on reports in the literature and bioinformatic analyses, and further experimental evidence is needed to determine whether these bacterial species provide a substantial growth increase for other prairie plant hosts.

## Perturbed root-endophytic microbial composition impacted plant performance in a species-specific manner

Anthropomorphic antibiotic usage has increased over the past few decades resulting in an accumulation of residues in multiple environmental areas such as manure and agricultural soils [46]. Antibiotics are well known to change the composition of the soil microbial community [98–100] as was reported in our study. However, this change in root-endophytic bacterial community did not result to any changes in plant performance. This could suggest that our antibiotic treatment did not adequately change the fungi:bacteria ratio by inhibiting bacterial species [98]. An alternative hypothesis could be that the root-endophytic bacterial community retained enough functional redundancy and diversity to withstand this perturbation. Follow-up work such as varying concentrations of the antibiotics is needed to determine the direct effect of antibiotics on root-endophytic bacterial communities.

We observed that with our second perturbation treatment, autoclaving soil, drastically reduced the diversity and changed the composition of the root-endophytic bacterial community (S1A and S2D Figs). Additionally, we did not see a reduction in plant performance for *C. nutans* and *C. canadensis* while we did for *H. helianthoides*, *M. fistulosa* and *R. pinnata* (Fig 4A). This result suggests that certain plant species would become dominant while others would become rare within the population after a prescribed fire. Furthermore, after the extreme perturbation, plant identity still explained most of the variation (23%) for the structure of the root-endophytic bacterial community (S2C Fig). This corroborates the theory that selection by plant hosts are largely driven by plant exudates that are recognized by the soil bacteria [101]. Additional studies are needed to understand the impact of current prairie restoration practices on root-endophytic bacterial communities and its impact on prairie community composition.

## Plant-soil feedbacks are not facilitated by the root-endophytic bacterial community

Plant-soil feedback studies have been used for years to elucidate the impact of the soil microbial community on plant performance [24, 27, 54, 102]. Many plant-soil feedback studies hypothesize that invasive plants exhibit strong positive feedbacks because the soil community does not harbor specialized pathogens for invasive plants and native plants exhibit strong negative feedbacks due to the soil community containing an accumulation of soil borne specialized pathogens [23, 26, 102]. We took advantage of plant-soil feedback experimental design and instead of attributing the effect to the entire community, we focused on the root-endophytic bacterial community to begin to understand the relationship between the root-endophytic bacterial community and plant performance.

In contrast to other studies, we did not detect a significant main effect plant-soil feedback (feedback between all plant hosts) within our experiment (S5 Fig); however, we did detect pair-wise plant-soil feedbacks between pairs of plant hosts (Fig 6). The measured performance of three of the native plant hosts suggests that the soil history independent of host source may have accumulated beneficial microbes rather than soil borne pathogens. As for the weedy plant host, *C. nutans*, performance was the same in conspecific and heterospecific soils and in field and autoclaved field soil (S4 Fig). We observed the complete opposite for two of our native plant hosts, where *M. fistulosa* had higher performance in conspecific rather than heterospecific soils and *C. canadensis* performed worse in conspecific rather than heterospecific soils (S4 Fig). Since plant-soil feedback is an interaction, we calculated the pair-wise interactions between each pair and discovered 4/10 significant (Fig 6A, 6C, 6E, 6I) and 2/10 (Fig 6G and 6K) closely significant feedbacks. However, only one (Fig 6J) had a significant correlation between the root-endophytic bacterial community composition and difference in plant performance. This suggests that the root-endophytic bacterial community is not facilitating plant-soil feedbacks. We have presented evidence based on bioinformatic analyses and an experimental design which included soil collected from fields, further experimental evidence in which soils are manipulated or trained by plant host prior to the experimentation is needed to determine the direct effect of host manipulation on the composition of root-endophytic bacterial communities.

## Conclusion

Our study advances the emerging field of plant-microbe interactions by showing that prairie endophytic root bacterial communities are structured by the plant host, regardless of perturbation to the soil community. Additionally, our study reveals a previously unknown correlation between the composition of the endophytic bacterial root community and plant biomass. Together, this study suggests that the composition of the root endophytic bacterial community could play an underestimated role in determining plant diversity and performance, and the stability of plant communities in response to environmental change. Further research aimed at detangling the direct effect of the composition of the root endophytic bacterial community on plant biomass is warranted.

## Supporting information

**S1 Fig. Reduction of root endophytic bacteria and total plant biomass in the autoclaved treated soil.** (a) Box-plots of alpha diversity (species richness) in the soils treated with antibiotics (yellow) and without antibiotics (brown). (b) Total plant biomass was unchanged due to antibiotic (yellow) treatment.
(EPS)

**S2 Fig.** (a) Post sonication and post wash bacterial communities are different from the bacteria found in the endophytic compartment. CAP analysis showing the contribution of location to overall composition. Ordination of Bray-Cutis dissimilarities shows clustering of root endophytic bacterial communities by location: after sonication (red), after wash (blue) and endophytic (green). (b) The source of the autoclaved field soil had no effect on total biomass averaged across all plant species in each soil inoculum [ANOVA p = 1] (c) the CAP analysis of Bray-Cutis dissimilarities shows clustering of root endophytic bacterial communities by plant host: *Carduus nutans* (red), *Conyza canadensis* (blue), *Heliopsis helianthoides* (green), *Monarda fistulosa* (purple), *Ratibida pinnata* (orange). [ADONIS p < 0.001, species—$r^2$ = 0.236, soil $r^2$ = 0.11)] (d) CAP plot showing both live and autoclaved soil endophytic root microbiome separated by soil treatment (field soil vs autoclaved field soil) [ADONIS p< 0.003,

$r^2 = 0.048$, n = 247].
(EPS)

**S3 Fig. OTUs that are differentially abundant in endophytic compartment of plant hosts—Box plots of differentially abundant OTUs between plant hosts produced by 'ANCOM' v1.1.3:** *Carduus nutans* **(red),** *Conyza canadensis* **(blue),** *Heliopsis helianthoides* **(green),** *Monarda fistulosa* **(purple),** *Ratibida pinnata* **(orange).**
(EPS)

**S4 Fig. Biomass is greatly reduced in natives (***H. helianthoides, M. fistulosa, R. pinnata***) prairie plant hosts compared to invasive (***C. nutans***) prairie plant hosts.** Total dried biomass weight for each plant host by field conspecific field soil (soil collected from host species) and heterospecific field soil (soil collected from other host species). The colors represent the soil condition: Field Soil (pink) and Autoclaved Field Soil (grey).
(EPS)

**S5 Fig. Performance of plant hosts of both target species and heterospecific hosts when grown with soil inoculum collected from target host vs heterospecific hosts.** The resulting interaction between host species and soil inoculum was used to define the strength and direction of feedback. A-E are the individual target species. F is the strength of feedback calculated using A-E.
(EPS)

**S6 Fig. Distribution of bacterial phyla in the endophytic compartment.** Bar plots of all classified endophytic bacterial OTUs separated by phyla and plant host. Bar plot show a dominance of Proteobacteria across all plant hosts.
(EPS)

**S1 Table. ADONIS of endophytic bacterial communities.**
(DOCX)

**S2 Table. ANOVA of total dried biomass.**
(DOCX)

**S3 Table. Pair-wise T-tests P-values for differentially abundant OTUs.**
(XLSX)

**S4 Table. Tukey's *post hoc* total dried biomass for soil inoculum.**
(DOCX)

**S5 Table. Taxonomy of OTUs.**
(XLSX)

**S6 Table. ANOVA using permutation tests of total dried biomass—Field soil vs autoclaved field soil.**
(DOCX)

**S7 Table. ANOVA using permutation tests of total dried biomass—Conspecific vs heterospecific.**
(DOCX)

## Acknowledgments

We thank Mike Dryer, Darlene Branson, Khadine Gomez, Max Herzog, Kevin Stiles, and the undergraduates of the Mangan Lab for their assistance in the greenhouse, Jessica Hoisington-

Lopez, Eric Martin, and Brian Koebbe for next-generation sequencing and high-throughput computing support at the Edison Family Center for Genome Sciences and Systems Biology at Washington University in St Louis School of Medicine, and members of the Dantas lab for insightful discussions of the results and conclusions.

## Author Contributions

**Conceptualization:** Boahemaa Adu-Oppong, Gautam Dantas.

**Data curation:** Boahemaa Adu-Oppong.

**Formal analysis:** Boahemaa Adu-Oppong, Scott A. Mangan.

**Funding acquisition:** Boahemaa Adu-Oppong, Scott A. Mangan, Gautam Dantas.

**Investigation:** Boahemaa Adu-Oppong, Scott A. Mangan, Claudia Stein.

**Methodology:** Boahemaa Adu-Oppong, Scott A. Mangan.

**Project administration:** Boahemaa Adu-Oppong, Gautam Dantas.

**Resources:** Boahemaa Adu-Oppong.

**Software:** Boahemaa Adu-Oppong.

**Supervision:** Boahemaa Adu-Oppong, Scott A. Mangan, Gautam Dantas.

**Validation:** Boahemaa Adu-Oppong.

**Visualization:** Boahemaa Adu-Oppong.

**Writing – original draft:** Boahemaa Adu-Oppong.

**Writing – review & editing:** Boahemaa Adu-Oppong, Scott A. Mangan, Claudia Stein, Christopher P. Catano, Jonathan A. Myers, Gautam Dantas.

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
