## [Decision Letter · Decision Letter 0]

9 Dec 2019

PONE-D-19-22834

Prairie plants harbor distinct and beneficial endophytic bacterial communities

PLOS ONE

Dear Dr. Adu-Oppong,

Thank you for submitting your manuscript to PLOS ONE. After careful consideration, we feel that it has merit but does not fully meet PLOS ONE’s publication criteria as it currently stands. Therefore, we invite you to submit a revised version of the manuscript that addresses the points raised during the review process.

Both reviewers and myself can see the excellent value in the work that you and your colleagues have reported in this manuscript.  However, both reviewers have done a thorough job in studying the details of the data and its presentation. If you elect to revise the manuscript, I will be looking to see that the reviewer's comments have been dealt with, either by revision or explanation.

We would appreciate receiving your revised manuscript by Jan 23 2020 11:59PM. To enhance the reproducibility of your results, we recommend that if applicable you deposit your laboratory protocols in protocols.io, where a protocol can be assigned its own identifier (DOI) such that it can be cited independently in the future. For instructions see: http://journals.plos.org/plosone/s/submission-guidelines#loc-laboratory-protocols

We look forward to receiving your revised manuscript.

Kind regards,

Ulrich Melcher

Academic Editor

PLOS ONE

Journal Requirements:

1. We note that you have stated that you will provide repository information for your data at acceptance. Should your manuscript be accepted for publication, we will hold it until you provide the relevant accession numbers or DOIs necessary to access your data. If you wish to make changes to your Data Availability statement, please describe these changes in your cover letter and we will update your Data Availability statement to reflect the information you provide.

2. Please amend either the title on the online submission form (via Edit Submission) or the title in the manuscript so that they are identical.

Reviewers' comments:

Reviewer's Responses to Questions

**Comments to the Author**

1. Is the manuscript technically sound, and do the data support the conclusions?

Reviewer #1: No

Reviewer #2: Yes

2. Has the statistical analysis been performed appropriately and rigorously? 

Reviewer #1: N/A

Reviewer #2: I Don't Know

3. Have the authors made all data underlying the findings in their manuscript fully available?

Reviewer #1: No

Reviewer #2: Yes

4. Is the manuscript presented in an intelligible fashion and written in standard English?

Reviewer #1: No

Reviewer #2: Yes

5. Review Comments to the Author

Reviewer #1: The manuscript “Prairie plants harbor distinct and beneficial root-endophytic bacterial communities” by Adu-oppong et al. focuses on the effects of endophytic bacterial communities in the roots of five prairie plants in their performance. The authors aimed to address compositional variations in bacterial community with respect to the host plants, stability of the bacterial community under disturbance and the overall fitness of the plants. The research questions are valid. The choice of native plants in the study and the experimental designs are commendable.

I have following comments for considerations.

The authors extensively used the term “plant-soil feedbacks” throughout the paper. It is described in the introduction as the feedbacks that influence plant evolution and diversity (line 94-96) as a response from plant microbiome but in the result and discussion sections, the term is used loosely. It creates confusion to the readers. A supplemental figure (S1) associated seems unnecessary and not sure if the authors have permission from the original authors to include the figure.

In figure S2, the alpha diversity measures for both antibiotic and non-antibiotic treated field soil and autoclaved field soil have overlapping quartile range, hence the alpha diversity measures may not be significant statistically unlike what is state in lines 187-188.

Not sure what authors wants to say in lines 189-191. The author mentioned they calculated the strength of the deterministic factors, which is not clear how they did.

In line 222, the equation for the plant-soil feedback interaction is given, however in any results no data based on the calculations were presented. In addition, based on the equation it seems that the interacting species always have antagonistic effects on each other. However, it is possible to have synergistic effects in nature.

Under characterizing root endophytic bacterial communities section, three different methods were mentioned. Not sure why different methods were adopted. The descriptions were difficult to follow. I did not find any descriptions how bacterial communities on the surface of the roots were avoided.

The sentence in line 281-283 is confusing. How the influence on plant diversity is measured by plant performance?

Figures like Figure 2A in line 309 and Figure 3B in line 310 were described together in the text while they were presented as sub-figures of two different main figures. It is inconvenient to refer the figures under two different main figures while in the text they are described together.

In Lines 313-317, the data mentioned about 100 OTUs were not supported by Fig 2B referred. In the figure, the abundance values of bacterial phylum across plant hosts did not seem different and might not be statistically significant. Also, in S6 Figure, I did not find any OTUs that was exclusively found in one plant. It was the abundance of some OTUs that differed widely among plants but none of the example was found with 13% OTUs exclusively found in one plant.

The description of abundance for three bacterial families in Lines 350-355 based on Figure 3C was not convincing, particularly for family Cytophagaceae, log abundance values for both C.canandensis and H. helianthoides were higher. Also, the weak effect of soil inoculum on plant phenotypes was mentioned in the description but throughout the manuscript the authors did not describe what host phenotypes they measured.

In Lines 361-363, it was inferred that the change in the plant biomass was due to the soil biotic components. When a soil is autoclaved, microorganisms are most likely killed but the abiotic components of the soil may remain intact, for example if some soils have high nutrient or organic content than the effect of such abiotic components could still be there after autoclaving.

The evidence presented in line 420-423 that the root endophytic bacterial community in autoclave soils were clustered by plant hosts was not very strong, particularly C. canadensis, H. helianthoides and R. pinnata in CAP biplot (S7B) did not show clear clustering.

Overall, many descriptions and results presented need to be revised for clarity. Many figures require proper rearrangement to make them convenient to refer while reading the text. I recommend detail revision of the manuscript before publication.

Reviewer #2: General comments:

This study explores the ecological significance of bacteria in the roots of plants. The study is well designed and addresses novel questions. I do think it has important information – as the authors state, this paper is starting to help understand the “black box” of soil microbes. I like the overall thrust of the paper with information on both plant performance and microbial communities.

However, I did find the paper somewhat hard to follow. It is a challenging topic to write about, because the reader needs to understand so many different things – i.e., the conceptual ideas of soil feedback, but also the details of how this specific study was done, both in the greenhouse and the molecular work. I have outlined several of these issues below in the specific comments.

Specific comments:

INTRODUCTION

The goal of the introduction, I think, is to emphasize what we do know and what we don’t know. I thought the first two paragraphs are useful and straightforward. The third paragraph (starting on 77) starts by stating that are few perturbations and that most studies focus on stressors. To me these are two separate ideas and I’m not sure why they are linked here? I think the key point of this third paragraph is to get readers thinking about antibiotics as a stressor that hasn’t been studied?

The paragraph starting on 92 emphasizes plant soil feedbacks. This is an important area and I am familiar with the literature, but for readers that are new to this concept, it is a bit tricky that the phrase is used on line 92 but not explained for a few more lines. As I will note later, I think more information about the general design of feedback studies needs to be introduced so that the reader can follow the methods.

I like that the introduction of the paper was written without regard to specifics of the study organisms (i.e., prairie plants and their microbes). However, the paragraph starting at 109 uses the word “prairie” several times without any explanation of this ecosystem. I am personally familiar with prairies but for an international journal, there needs to be some ecological context provided. (A very minor point: on 110, I’d refer to native prairie “vegetation” or “community” or “ecosystem” rather than “population” – i.e., plant species cannot co-exist within populations).

MATERIALS AND METHODS

130 – typo

136 – I realize that the prairie restoration plots are simply the source of the soil and not a focus of the study being presented here. However, a little more information about the plots would be useful (how many prairie species were used in the restoration plots?)

147-148 Since I am familiar with feedback studies, I know exactly what is meant by this sentence and its reference to conditioning. However, many readers would not be clear on this. I think the soil feedback paragraph in the introduction has to be rewritten to emphasize the two stage process – conditioning and testing, so that readers can understand what you mean by mimicking the first stage by using soil collected from field grown plants.

152 – Minor point – I’m not sure about the meaning of this sentence.

155-6 – I’m guessing that the bulk soil noted here may be the same thing as the field soil noted in line 164, but most readers wouldn’t easily follow this. Again, I think it is important to write this paper without expecting that readers already have done plant feedback studies. For example, when one reads lines 155-156, there is no sense of why the bulk soil was collected (it may be better to refer to this bulk soil collection when you are talking about creating the background soil).

Figure 1 is useful. However, I can’t see the color difference for the water/antibiotic treatments.

Just FYI, I found it distracting to have the figure legends embedded in the middle of the text (but without the figures). I think most reviewers expect the figure legends to be at the end of the manuscript, right before the figures.

186 I found it initially odd to have a section on autoclaving and antibiotics after these terms had already been used in the previous section. Perhaps it would be useful to have the section heading for 157 be listed as “Overview of the Greenhouse Experiment” so readers may realize that other sections will provide many of the details.

188, 194, 195 It seems that the supplementary figures noted here are providing results from this study? Seems rather odd to have results presented in the methods section of the paper?

212,213 I didn’t see reference to the size of the pots? There is reference to “optimal time for growth” but I’m not sure what that means. The plant species used in this study can become very large in field environments. I realize that a greenhouse study by definition has to deal with smaller plants, but some reference to the capacity of the pots is important to consider. Did the plants get “root-bound”?

219 I am familiar with this equation, but what I don’t see in this section is exactly how to apply it to the data being collected. I would add a few more sentences so readers can understand how the biomass data collected from the different pots can be used to determine the alpha and beta values needed for the equation.

226 A general point. After reading the paper a couple of times, I have understood that the root endophytic bacterial communities were “created/manipulated” by taking sterile plants and exposing them to different soils. One could then characterize them by looking at the bacterial communities within the root samples using sterile techniques. I think it might be useful to readers if these general points were made early in the paper just so that all readers quickly see the big picture.

244 something is mixed up in this sentence

266 very minor point – data “is” plural so should use “were” here

285 I assume soil history means what plant the soil was collected under? Perhaps I missed it, but I would make sure this phrase was defined earlier.

287 A minor point, but “acquisition” seems a bit of an odd word. Isn’t autoclaving killing bacteria so that they never enter the plant? I might of used something more general like “the characteristics of root endophytic bacterial communities……….”

309, 310 I looked at figure 2a and 3b and can see the point the authors are making. However, I find the statistics presented a bit confusing. Is the author meaning that the difference between r squared of 11 and .03 is justification for the statement? I also had a similar question in the next sentence where the difference in r squared is .07 versus .02 (lines 312 and 313). I guess I’m not completely sure what the p value and r squared are telling me and in particular whether comparison of r2 values that already are quite small is meaningful (but like I said, the Fig 2A and 3B was convincing so perhaps this is all right?)

334 I was unclear about this sentence – in the methods it implied that feedback was being measured? – there were no qualifications?

Results in Fig 4 are quite interesting!

413-415, 424-427 A comment about section headings. I was confused to see a subsection heading under a section heading for situations where there was only one subsection.

A general point: This paper had a very large number of supporting figures and tables. Most readers will not take the time to look at all of these. I think focusing on a small number of figures and tables would improve the message of the paper.

DISCUSSION

465 I am confused by the reference to selection here. In the introduction, there is reference to “selection” in the context of the influence of the plant host. In 465, I’m not sure what is being meant, especially since it seems that these results do show that “selection by the plant host” is occurring.

474 typo – I expect you meant “rather than artificially”

548 typo – “took advantage of “

6. PLOS authors have the option to publish the peer review history of their article (what does this mean?). If published, this will include your full peer review and any attached files.

Reviewer #1: No

Reviewer #2: No

---

## [Author Response · Author response to Decision Letter 0]

23 Jan 2020

Dear Ulrich Melcher, 

We submit for your consideration a revised version of manuscript PONE-D-19-22834 entitled “Prairie plants harbor distinct and beneficial root-endophytic bacterial communities” by Boahemaa Adu-Oppong, Scott A. Mangan, Claudia Stein, Christopher P. Catano, Jonathan A. Myers, and Gautam Dantas. 

We thank the editor and the reviewers for their time and critical assessment of our manuscript and thoughtful suggestions to help improve our manuscript. A common major concern was to improve clarity of our statistical analyses and reduce the number of supplemental figures. Our revised manuscript includes substantive textual edits and revisions to main text figures 1, 2, and 6, and supplemental figure 6. We have reduced the number of supporting figures to 6 from 10. Having addressed all reviewer concerns, our revised manuscript is greatly strengthened.

Again, we thank the reviewers for their careful comments and feel our paper has been vastly strengthened and clarified by addressing them. The following pages include our responses to the specific comments of the reviewers and the reviewer comments in italics.

Sincerely, 

Boahemaa Adu-Oppong, Ph.D., on behalf of all the authors

 

1st reviewer comment’s addressed: 

The authors extensively used the term “plant-soil feedbacks” throughout the paper. It is described in the introduction as the feedbacks that influence plant evolution and diversity (line 94-96) as a response from plant microbiome but in the result and discussion sections, the term is used loosely. It creates confusion to the readers. A supplemental figure (S1) associated seems unnecessary and not sure if the authors have permission from the original authors to include the figure.

Thank you for the feedback regarding our explanation of plant-soil feedbacks. We have removed the supplemental figure since it was deemed unnecessary and have included more information in the introduction to help tie the results back to the idea behind plant-soil feedbacks. 

We included these sentences in the introduction:

The term feedback within the scope of this study involves 2 steps: 1) the plant host perturbs the composition of the bacterial community, and 2) this differentiation must affect the performance of the plant host{Bever, 2003 #6}. (Lines 93 – 95)

We included these sentences to the methods since we went into detail on how to do steps 1 and 2 which wasn’t suitable for the introduction: 

To create differentiated soil communities (step 1), one can either allow plant hosts to grow in similar initial soil communities for a few months or sample close to adult plants due to the short generation time and rapid community dynamics of microbial communities {Bever, 2010 #435}. To measure the performance of the plant host (step 2), the plants are then grown in an inoculation of the differentiated soil communities surrounded by a common background soil to isolate microbial effects {Bever, 2010 #435}. We chose to collect soil from these experimental plots to serve as our differentiated soils, which we will refer to as soil history throughout the manuscript. (Lines 149 – 155)

In figure S2, the alpha diversity measures for both antibiotic and non-antibiotic treated field soil and autoclaved field soil have overlapping quartile range, hence the alpha diversity measures may not be significant statistically unlike what is state in lines 187-188.

Thank you for the feedback on the figure S2. We used the aovp function in the lmPerm R package which performs a Fitting and testing ANOVA using permutation test. The reported p value is <2e-16 showing that the alpha diversity is different between the field soil and the autoclaved field soil. We have included the results in S1 Table. Additionally, added these lines to the manuscript: 

The aovp function in the lmPerm package which performs a fitting and testing ANOVA using permutation test was used to determine differences between alpha diversity of the root endophytic bacterial communities and total dried biomass between the plant hosts. (Lines 294-296)

Not sure what authors wants to say in lines 189-191. The author mentioned they calculated the strength of the deterministic factors, which is not clear how they did.

We have removed this sentence and went into detail about how we calculated the strength of the deterministic factors using the ADONIS r 2 value. We added these lines to the manuscript: 

The r2 value from the ADONIS reflects the amount of variation in microbial community composition explained by each of the factors tested. We corrected for multiple comparisons with the False Discovery Rate post-hoc test to determine which pairs were significantly different. (Lines 290 – 293)

In line 222, the equation for the plant-soil feedback interaction is given, however in any results no data based on the calculations were presented. In addition, based on the equation it seems that the interacting species always have antagonistic effects on each other. However, it is possible to have synergistic effects in nature.

The panels within each of the figures displaying the strength of feedback is calculated based on the equation in 222. Additionally, in Figure 6, we show that there can be synergistic effects as well. To make this link clearer, we added a few sentences describing how we determined if the interactions were significant and changed the y-axis legend on Figure 6 to Is (plant-soil feedback interaction strength). 

Lines added: 

We used a one-sample t-test to determine if the feedback interaction coefficient was significantly different from 0. (Lines 239 – 241)

Under characterizing root endophytic bacterial communities section, three different methods were mentioned. Not sure why different methods were adopted. The descriptions were difficult to follow. I did not find any descriptions how bacterial communities on the surface of the roots were avoided.

The after sonication is the step that was used at the onset of prepping the samples for bacterial DNA extraction which first removes bacteria on the surface of the roots. Afterwards, the samples where washed and centrifuged to further remove any other bacteria that remained. To make this a bit clearer, we included our reasonings for doing the sonication and the washes. This was added to the manuscript: 

We sonicated and washed the roots to remove any bacteria that could be found on the external surface of the roots. The roots were aseptically transferred to a new 15ml and freeze dried overnight. We validated that the sonication and washes were sufficient in removing surface and exterior bacteria by sequencing these extracts and determining that the microbial communities were significantly different (S2A Fig). (Lines 259 – 263)

The sentence in line 281-283 is confusing. How the influence on plant diversity is measured by plant performance?

Thank you for bringing this to our attention. We tried to make the link less confusing by using terminology we have used throughout the manuscript. We want to measure the direct link between the microbial community and plant biomass that could lead to the differences seen when calculating plant-soil feedbacks which is a measure of plant biomass grown in different microbial communities (soil inoculum). This is our new sentences: 

Measuring plant-soil feedbacks only involve measuring biomass and manipulating the soil inoculum (soil history). We correlated the composition of the bacterial communities to plant performance to understand if differences in endophytic root bacterial compositions could explain differences in plant biomass. (Lines 305-308)

Figures like Figure 2A in line 309 and Figure 3B in line 310 were described together in the text while they were presented as sub-figures of two different main figures. It is inconvenient to refer the figures under two different main figures while in the text they are described together.

Thank you for pointing this out. We did this to make the conclusions drawn from the figures less confusing to the reader. In Figure 2, we are only concerned about the plant species while in Figure 3 we are only concerned about the soil inoculum which were soils collected from the different plant species. Since the colors refer to the plant species, we did not want readers to be confused about the legend. We have decided to recreate Figure 2 to be a combination of previous Figure2A and Figure 3B and in the legend noted that the colorings either refer to the soil history or the plant host. 

In Lines 313-317, the data mentioned about 100 OTUs were not supported by Fig 2B referred. In the figure, the abundance values of bacterial phylum across plant hosts did not seem different and might not be statistically significant. Also, in S6 Figure, I did not find any OTUs that was exclusively found in one plant. It was the abundance of some OTUs that differed widely among plants but none of the example was found with 13% OTUs exclusively found in one plant.

Thank you for raising this concern. Figure 2B was meant to be a representation of the phylum not OTUs since that figure would be hard to understand with the increase number of colors within the stack bar plot. We have decided to remove Figure 2B since this figure was deemed not useful. Figure S6 is a bit hard to read since we wanted to plot all 100 OTUs which where labeled as significantly differently abundant using the ANCOM statistical package (https://www.ncbi.nlm.nih.gov/pubmed/26028277). The OTUs which were statistically significantly found in one plant and not the other 4 are: OTU 109, OTU 193, OTU 197, OTU 261, OTU 344, OTU 379, OTU 415, OTU 45, OTU 50, OTU 301, OTU 229, OTU 168, OTU 179. Here is a recent paper that have made use of the same statistical package (https://journals.plos.org/plosone/article?id=10.1371/journal.pone.0225079). To make this clearer, we performed a similar pairwise t-tests and have included boxes around those particular OTUs and included the statistical test as a new supplemental table 3. The explanation for the statistical tests were added to the manuscript: 

We then conducted a pairwise-t-test with Bonferroni adjusted p-values to determine which pairs where significantly different. (Lines 301-303)

The description of abundance for three bacterial families in Lines 350-355 based on Figure 3C was not convincing, particularly for family Cytophagaceae, log abundance values for both C.canandensis and H. helianthoides were higher. Also, the weak effect of soil inoculum on plant phenotypes was mentioned in the description but throughout the manuscript the authors did not describe what host phenotypes they measured.

Thank you for raising this concern. The statistical test used was called ANCOM (https://www.ncbi.nlm.nih.gov/pubmed/26028277) which is a statistical package designed specifically for microbiome communities which are usually zero-inflated gaussian distributed. The OTUs which were differentially abundant are not exclusively found in one soil type but rather the abundance is different between the 5. Therefore, 2 can share the same abundance and be different in the other 3. It is like an ANOVA, which you have to perform a Tukey-post hoc to discover which ones are different from the others. Here is a recent paper that included a figure which used results from ANCOM as well (https://journals.plos.org/plosone/article?id=10.1371/journal.pone.0225079). To make this clearer, we performed a similar pairwise t-tests to determine which soil history harbored different amounts of the OTU. The term phenotype has been changed to plant host. We did not measure any host phenotypes within the scope of this study. Therefore, we have gone through and made sure that we did not use the term plant phenotype anywhere else in the manuscript. 

The explanation of the stats: 

We then conducted a pairwise-t-test with Bonferroni adjusted p-values to determine which pairs where significantly different. (Lines 301-303)

In Lines 361-363, it was inferred that the change in the plant biomass was due to the soil biotic components. When a soil is autoclaved, microorganisms are most likely killed but the abiotic components of the soil may remain intact, for example if some soils have high nutrient or organic content than the effect of such abiotic components could still be there after autoclaving.

We agree with the reviewer that if the soil had high nutrient effect then it should be there after autoclaving. We minimized the effect of this by using an inoculum (6% of the pot volume) which was described in line 174. If the abiotic component was higher in one soil source inoculum and influenced plant biomass, we would see that captured in Supplementary Figure 7. 

The evidence presented in line 420-423 that the root endophytic bacterial community in autoclave soils were clustered by plant hosts was not very strong, particularly C. canadensis, H. helianthoides and R. pinnata in CAP biplot (S7B) did not show clear clustering.

Yes, we agree that the clustering isn’t as clear as it could be, but we included the ADONIS as a measure of statistical differences between the microbial communities (ADONIS p < 0.001) and plant host captured 24% of the variation. 

2nd Reviewer comment’s addressed: 

Specific comments:

INTRODUCTION

The third paragraph (starting on 77) starts by stating that are few perturbations and that most studies focus on stressors. To me these are two separate ideas and I’m not sure why they are linked here? I think the key point of this third paragraph is to get readers thinking about antibiotics as a stressor that hasn’t been studied?

Yes, we agree with the second reviewer that the third paragraph was intended to have readers think about our antibiotic treatment as a stressor that hasn’t been studied. The nuance is that many studies focus on perturbing the plant community rather than perturbing the microbial community. To make sure that point is clear, we have restructured the third paragraph and can find the new start of the paragraph below: 

Second, few studies perturb the root-endophytic bacterial community with mechanisms which drastically impact the composition to understand the functional relationship of the plant microbiome (34-37), instead many focus on perturbations that directly affect plant community dynamics. (Lines 71-74)

The paragraph starting on 92 emphasizes plant soil feedbacks. This is an important area and I am familiar with the literature, but for readers that are new to this concept, it is a bit tricky that the phrase is used on line 92 but not explained for a few more lines. As I will note later, I think more information about the general design of feedback studies needs to be introduced so that the reader can follow the methods.

Thank you for the insightful feedback. We have included more information in the introduction to describe plant-soil feedback study design and also added a few sentences to the method section. Below are the sentences added to the introduction: 

This interaction is characterized as the plant-soil feedback framework due to observations in which plant species differ in their response to individual microbial species and in turn, growth rates of individual microbial species are also affected by the plant host (59). The term feedback within the scope of this study involves 2 steps: 1) the plant host perturbs the composition of the bacterial community, and 2) this differentiation must affect the performance of the plant host{Bever, 2003 #6}. (Lines 90 – 95)

The below lines were added to the methods section: 

To create differentiated soil communities (step 1), one can either allow plant hosts to grow in similar initial soil communities for a few months or sample close to adult plants due to the short generation time and rapid community dynamics of microbial communities {Bever, 2010 #435}. To measure the performance of the plant host (step 2), the plants are then grown in an inoculation of the differentiated soil communities surrounded by a common background soil to isolate microbial effects {Bever, 2010 #435}. We chose to collect soil from these experimental plots to serve as our differentiated soils, which we will refer to as soil history throughout the manuscript. (Lines 149 – 155)

I like that the introduction of the paper was written without regard to specifics of the study organisms (i.e., prairie plants and their microbes). However, the paragraph starting at 109 uses the word “prairie” several times without any explanation of this ecosystem. I am personally familiar with prairies but for an international journal, there needs to be some ecological context provided. (A very minor point: on 110, I’d refer to native prairie “vegetation” or “community” or “ecosystem” rather than “population” – i.e., plant species cannot co-exist within populations).

Thank you very much for bringing up the concerns about introducing the idea of a prairie to an international audience. We have provided ecological context and those lines are found below: 

The North American prairie ecosystem is one of the Earth’s most endangered ecosystem and only 13% of its native range remains intact {Samson, 1996 #193}. (Lines 108-09)

Additionally we replaced the word populations to ecosystem in line 106.

MATERIALS AND METHODS

130 – typo

Fixed typo. (Horsweeed) -> Horseweed on line 129

136 – I realize that the prairie restoration plots are simply the source of the soil and not a focus of the study being presented here. However, a little more information about the plots would be useful (how many prairie species were used in the restoration plots?)

We have added more information about the plots can be found in the recent publication. The below lines have been added to the manuscript: 

The plots were seeded with 25 Missouri ecotype native forb and 5 grass species, the seeding densities and common names can be found here along with more information about the experimental prairie restoration plots:{Wohlwend, 2019 #634}. (Lines 145 – 148) 

147-148 Since I am familiar with feedback studies, I know exactly what is meant by this sentence and its reference to conditioning. However, many readers would not be clear on this. I think the soil feedback paragraph in the introduction has to be rewritten to emphasize the two stage process – conditioning and testing, so that readers can understand what you mean by mimicking the first stage by using soil collected from field grown plants.

Thank you for this insightful feedback. The paragraph where we introduce soil-feedbacks has been re-written to include information about the 2-stage process. These changes are captured in the lines below: 

This interaction is characterized as the plant-soil feedback framework due to observations in which plant species differ in their response to individual microbial species and in turn, growth rates of individual microbial species are also affected by the plant host (59). The term feedback within the scope of this study involves 2 steps: 1) the plant host perturbs the composition of the bacterial community, and 2) this differentiation must affect the performance of the plant host{Bever, 2003 #6}. (Lines 90 – 95)

 The below lines were added to the methods section: 

To create differentiated soil communities (step 1), one can either allow plant hosts to grow in similar initial soil communities for a few months or sample close to adult plants due to the short generation time and rapid community dynamics of microbial communities {Bever, 2010 #435}. To measure the performance of the plant host (step 2), the plants are then grown in an inoculation of the differentiated soil communities surrounded by a common background soil to isolate microbial effects {Bever, 2010 #435}. We chose to collect soil from these experimental plots to serve as our differentiated soils, which we will refer to as soil history throughout the manuscript. (Lines 149 – 155)

152 – Minor point – I’m not sure about the meaning of this sentence.

The structure of the sentence was not correct. We have moved the date of the collection of soils to the beginning of the line 157.

155-6 – I’m guessing that the bulk soil noted here may be the same thing as the field soil noted in line 164, but most readers wouldn’t easily follow this. Again, I think it is important to write this paper without expecting that readers already have done plant feedback studies. For example, when one reads lines 155-156, there is no sense of why the bulk soil was collected (it may be better to refer to this bulk soil collection when you are talking about creating the background soil).

Thank you for bringing this to our attention. We have changed the syntax on line 164-165 to make sure the readers can follow the methods and understand the experimental design from the introduction into the methods section. The added lines are below: 

Bulk soil was collected 30m away from the experimental Tyson plots to serves as our common background soil. All Soils were stored in the dark at 4℃.

Figure 1 is useful. However, I can’t see the color difference for the water/antibiotic treatments.

Thank you for finding our figure useful. We have changed the colors so you can see the difference between the water/antibiotic treatments. 

Just FYI, I found it distracting to have the figure legends embedded in the middle of the text (but without the figures). I think most reviewers expect the figure legends to be at the end of the manuscript, right before the figures.

Thank you for bringing up the point about figure legends. These are the instructions for PLOS ONE: Figure captions must be inserted in the text of the manuscript, immediately following the paragraph in which the figure is first cited (read order). Do not include captions as part of the figure files themselves or submit them in a separate document.

186 I found it initially odd to have a section on autoclaving and antibiotics after these terms had already been used in the previous section. Perhaps it would be useful to have the section heading for 157 be listed as “Overview of the Greenhouse Experiment” so readers may realize that other sections will provide many of the details.

Thank you for the insightful comment. We have changed the section heading to “Overview of the Greenhouse Experiment” on line 166.

188, 194, 195 It seems that the supplementary figures noted here are providing results from this study? Seems rather odd to have results presented in the methods section of the paper?

We used supplementary figures within the methods section to demonstrate the reasoning behind how we constructed the study that didn’t directly answer our main hypotheses. Other papers have done this before such as: https://journals.plos.org/plosone/article?id=10.1371/journal.pone.0226432, https://journals.plos.org/plosone/article?id=10.1371/journal.pone.0200974, https://journals.plos.org/plosone/article?id=10.1371/journal.pone.0206484). 

212,213 I didn’t see reference to the size of the pots? There is reference to “optimal time for growth” but I’m not sure what that means. The plant species used in this study can become very large in field environments. I realize that a greenhouse study by definition has to deal with smaller plants, but some reference to the capacity of the pots is important to consider. Did the plants get “root-bound”?

We agree with the reviewer that the plant species chosen for this study can get quite large in the field and we restricted the experiment to 4 months because 2 (Heliopsis helianthoides and Monarda fistulosa) of the 5 were starting to become “root-bound”. We will remove the term “optimal time for growth” as to not confuse readers and refer to the life cycle stage. 

The lines below have been added to the manuscript: 

The experimented started July 2013 and ended October 2013. The duration was chosen to ensure all plants had enough time within the vegetative stage. (Lines 219 -220)

 Also added the pot dimension which was 6 inches in diameter to line 174. 

219 I am familiar with this equation, but what I don’t see in this section is exactly how to apply it to the data being collected. I would add a few more sentences so readers can understand how the biomass data collected from the different pots can be used to determine the alpha and beta values needed for the equation.

Thank you for bringing this to our attention. We have included a few sentences to explain in further detail how we used the equation to calculate the feedback coefficient. We hope the example is easy to follow. The lines below were added to the manuscript: 

We demonstrate how we calculated the interaction coefficient using an example with H. helianthoides and C. nutans. Total dried biomass of H. helianthoides grown in inoculum collected from conspecifics = ∝_A. Total dried biomass of H. helianthoides grown in inoculum collected from C. nutans = β_A. Total dried biomass of C. nutans grown in inoculum from H. helainthodies = ∝_B. Our final value variable, β_B, is the total dried biomass of C.nutans grown in conspecific inoculum. (Lines 233 – 238)

226 A general point. After reading the paper a couple of times, I have understood that the root endophytic bacterial communities were “created/manipulated” by taking sterile plants and exposing them to different soils. One could then characterize them by looking at the bacterial communities within the root samples using sterile techniques. I think it might be useful to readers if these general points were made early in the paper just so that all readers quickly see the big picture.

Thank you for this comment. We have included the point that they were grown in sterile conditions to the statement in the end of the introduction. Please see line below:

To address these questions, we sequenced the 16S rRNA gene from the endophytic root compartment of plants which were initially grown in sterile conditions and compared the bacterial community composition to the data collected from the plant-soil feedback study. (Lines 120-123)

244 something is mixed up in this sentence

Thank you for noticing that mistake. We have rewritten this sentence and the lines below have been added to the manuscript: 

We validated that the sonication and washes were sufficient in removing surface and exterior bacteria by sequencing these extracts and determining that the microbial communities were significantly different (S2A Fig). (Lines 261 – 263)

266 very minor point – data “is” plural so should use “were” here

Thank you for bringing this to our attention. The new line is below:

Microbial community count data were transformed using the DESeq2 package in R based on previous recommendations (73, 74). (Lines 283 – 284)

285 I assume soil history means what plant the soil was collected under? Perhaps I missed it, but I would make sure this phrase was defined earlier.

Yes, that is correct. We have included this definition earlier in the manuscript. The new line added to the manuscript is below:

We chose to collect soil from these experimental plots to serve as our differentiated soils, which we will refer to as soil history throughout the manuscript. (Lines 154 – 155)

287 A minor point, but “acquisition” seems a bit of an odd word. Isn’t autoclaving killing bacteria so that they never enter the plant? I might of used something more general like “the characteristics of root endophytic bacterial communities……….”

Yes, we have changed the word “acquisition” to composition of root endophytic bacterial community. The new line is found below: 

To link performance to composition of root endophytic bacterial communities in perturbed states, we tested for an interaction between plant host and autoclaving of field soil and plant host and soil history. (Lines 312 – 315)

309, 310 I looked at figure 2a and 3b and can see the point the authors are making. However, I find the statistics presented a bit confusing. Is the author meaning that the difference between r squared of 11 and .03 is justification for the statement? I also had a similar question in the next sentence where the difference in r squared is .07 versus .02 (lines 312 and 313). I guess I’m not completely sure what the p value and r squared are telling me and in particular whether comparison of r2 values that already are quite small is meaningful (but like I said, the Fig 2A and 3B was convincing so perhaps this is all right?)

Yes, the reviewer is correct. We are comparing the r squared values from the ADONIS report as a way of comparing which main effect explains most of the variation between the endophytic-root bacterial communities. To make this a bit clearer, we have included the statement ‘indicated by the r2 value’ to help draw readers to the r2 values. 

The lines below were added: 

In a subset of samples which were non-autoclaved and non-antibiotic treated, the compositional differences in root bacterial community was explained by plant identity (ADONIS p < 0.001, r2=0.11, Fig 2A) than by soil history (ADONIS p < 0.001, r2=0.03, Fig 2B) which is indicated by the r2 value. (Lines 333 – 336)

334 I was unclear about this sentence – in the methods it implied that feedback was being measured? – there were no qualifications?

Thank you for bringing this to our attention. We meant to say that the main effects were not significant therefore, the statement has changed and the new edits can be found below: 

Even though we did not demonstrate significant main plant-soil feedbacks, we sought to determine if specific soil histories affected plant performance, we measured the dried biomass of all plants grown in each of the field-soil inocula. (Lines 356 – 358) 

413-415, 424-427 A comment about section headings. I was confused to see a subsection heading under a section heading for situations where there was only one subsection.

Thank you for bringing this to our attention. Our initial thoughts were to have the headings match the main hypotheses so that it would be easy for readers to follow. After receiving this feedback, we have decided to remove the section headings and replaced them with the subsection heading. 

A general point: This paper had a very large number of supporting figures and tables. Most readers will not take the time to look at all of these. I think focusing on a small number of figures and tables would improve the message of the paper.

Thank you for this comment. We have tried to decrease the number of supporting figures and tables. There were some information that reviewer 1 brought up that needed justification so we added some information to the supplement. We have reduced the number of supporting figures to 6 from 10. 

DISCUSSION

465 I am confused by the reference to selection here. In the introduction, there is reference to “selection” in the context of the influence of the plant host. In 465, I’m not sure what is being meant, especially since it seems that these results do show that “selection by the plant host” is occurring.

Thank you for bringing that to our attention. That was a typo and this has been corrected. Please see the new line below: 

While selection by the plant host have been proposed as mechanisms structuring the soil microbial community (33, 43, 83), we provide evidence that the root-endophytic bacterial community is largely structured by the plant host, regardless of variation in soil history. (Lines 491 – 493)

474 typo – I expect you meant “rather than artificially”

Yes, we did mean to include the word than. We have corrected that mistake and the new changes are found below: 

Within our study, we used natural variation within each plant host rather than artificially manipulating the plant genome and noticed large variation between individuals within each plant host indicating that root endophytic bacterial microbiome is not identical within all individuals of the same plant host; however, they are more similar to each other than to different plant hosts (Fig 2A). (Lines 499 – 503)

548 typo – “took advantage of “

Thank you for reporting this typo. We have corrected the mistake and the updated lines are found below: 

We took advantage of plant-soil feedback experimental design and instead of attributing the effect to the entire community, we focused on the root-endophytic bacterial community to begin to understand the relationship between the root-endophytic bacterial community and plant performance. (Lines 573 – 576)

---

## [Decision Letter · Decision Letter 1]

29 Jan 2020

PONE-D-19-22834R1

Prairie plants harbor distinct and beneficial root-endophytic bacterial communities

PLOS ONE

Dear Dr. Adu-Oppong,

Thank you for submitting your manuscript to PLOS ONE. After careful consideration, we feel that it has merit but does not fully meet PLOS ONE’s publication criteria as it currently stands. Therefore, we invite you to submit a revised version of the manuscript that addresses the points raised during the review process.

We would appreciate receiving your revised manuscript by Mar 14 2020 11:59PM. To enhance the reproducibility of your results, we recommend that if applicable you deposit your laboratory protocols in protocols.io, where a protocol can be assigned its own identifier (DOI) such that it can be cited independently in the future. For instructions see: http://journals.plos.org/plosone/s/submission-guidelines#loc-laboratory-protocols

We look forward to receiving your revised manuscript.

Kind regards,

Ulrich Melcher

Academic Editor

PLOS ONE

Reviewers' comments:

Reviewer's Responses to Questions

**Comments to the Author**

1. If the authors have adequately addressed your comments raised in a previous round of review and you feel that this manuscript is now acceptable for publication, you may indicate that here to bypass the “Comments to the Author” section, enter your conflict of interest statement in the “Confidential to Editor” section, and submit your "Accept" recommendation.

Reviewer #2: (No Response)

2. Is the manuscript technically sound, and do the data support the conclusions?

Reviewer #2: Yes

3. Has the statistical analysis been performed appropriately and rigorously? 

Reviewer #2: I Don't Know

4. Have the authors made all data underlying the findings in their manuscript fully available?

Reviewer #2: Yes

5. Is the manuscript presented in an intelligible fashion and written in standard English?

Reviewer #2: No

6. Review Comments to the Author

Reviewer #2: General comments

This rewritten paper has addressed many of my comments and the introduction and methods are much easier to read.

I especially appreciate the new sentences relating to the feedback framework in the introduction and methods. However, even though the paper is much improved with reference to feedback, there are a few places where I still found it could be improved. I think these comments are particularly important given that the first sentence of the abstract refers to feedback. For example:

Currently, early in the Results (line 356), the authors state “even though we did not demonstrate significant plant soil-feedback” but no analyses or information are presented. Readers have to wait until lines 455-465 to understand the details of the feedback results. The authors should think about the order in which they present material. If feedback is central to the paper, I would present this information early in the Results.

I also think lines 577 – 578 will be confusing to many (i.e., the distinction between the two ways of talking about feedback).

Many specific comments are noted below. In general I found the manuscript much improved, but there are some errors involving words or sentences, or awkward writing, that I believe can be easily addressed.

36 – 37 – suggest rewrite “; in contrast to the degree to which ……”

38 – I would have said “influences”

71 – suggest rewrite “relationships between the plant and the microbiome”

73 – long sentence – perhaps delete phrase after comma

78 – suggest rewrite “……….such as drought. This is important given that current predictions state that the intensity…………”

81 – suggest rewrite “………soil microbial diversity (as expected with antibiotic use) leads to……

81 – suggest rewrite “………..performance, and even fewer”

95 – I might delete “must” (I know this is a part of feedback, but sentence structure seems awkward with it present)

98 – add comma after herbivore

100 – add “plant” before the word “species”

109 – I haven’t read this book but 13% seems too high. Often one sees Samson and Knopf Bioscience 44:418-421 cited, with a reference to 1-4% prairies left. It is tricky – tables in this article show 1% prairie left in most states and more in a few states. These articles are also quite old – more prairie has been lost in the last 20 years. But alas, a really good reference doesn’t exist to my knowledge.

129 – need spaces between 1 and invasive

149 – suggest rewrite “(step 1 of the plant-feedback framework)”

150 - do you mean “sample closer to adult plants in field sites”??

160 – Is this right? I would have expected soils to be pooled across experimental plots for those collected under the same plant species?

201-2 - It is stated that plant performance was not affected when grown in field autoclaved or field soil, but I wasn’t sure whether this was a specific result of this study (if so, I’d expect a phrase like “See results” at the end) or whether this was a general statement (I’d expect to see a citation or at least xxx, unpublished data)

219 Typo – “experiment” not “experimental”

229 – suggest rewrite “above-ground plant parts”

238-239 - missing words in sentence?

248 – not sure of the meaning of root position

250 insert “the” before “portion”

263 insert “the” before “microbial community”

306 perhaps insert “additionally” before “correlated” if the goal is to show this is an additional strength of the study

308 insert “the” before “linear”

335 suggest rewrite “was better explained (as indicated by the r2 value) by plant identity…………..

357 there is a problem with the flow of this sentence

394 replace comma with semi colon (or more generally think about how to reduce the complexity and structure of this sentence)

407 – 410 I understand the meaning of the sentence but it is cumbersome to read.

445-446 Reword. The reference to “we perturb” makes it sound like a methods sentence but clearly it is a results sentence

503 – It seems odd to emphasize the idea that individual plants within the same species are different before addressing the larger point that different host species structure the microbial community

559 - should say “would become dominant.” Also, I think it is too much of a stretch to infer from a single lab study with autoclaving whether particular species would be more common after a fire. Many factors affect post-fire response.

7. PLOS authors have the option to publish the peer review history of their article (what does this mean?). If published, this will include your full peer review and any attached files.

Reviewer #2: No

---

## [Author Response · Author response to Decision Letter 1]

2 Apr 2020

Currently, early in the Results (line 356), the authors state “even though we did not demonstrate significant plant soil-feedback” but no analyses or information are presented. Readers have to wait until lines 455-465 to understand the details of the feedback results. The authors should think about the order in which they present material. If feedback is central to the paper, I would present this information early in the Results.

Thank you for bringing this to our attention. We have removed the comment about plant soil-feedbacks here since those specific results are not discussed until later (following a parallel structure based on the 4 questions we posed at the end of the introduction). 

The new line is below: 

To determine if specific soil histories affected plant performance, we measured the dried biomass of all plants grown in each of the field-soil inocula. (Lines 356-357)

I also think lines 577 – 578 will be confusing to many (i.e., the distinction between the two ways of talking about feedback).

Thank you for that insightful comment. We have added more context and the new line is below: 

In contrast to other studies, we did not detect a significant main effect plant-soil feedback (feedback between all plant hosts) within our experiment (S5 Fig); however, we did detect pair-wise plant-soil feedbacks between pairs of plant hosts (Fig 6). (Lines 578 - 580)

36 – 37 – suggest rewrite “; in contrast to the degree to which ……”

Following your suggestion we have re-written the sentence: 

“Most studies focus on the microbial community in the soil rhizosphere; therefore, the degree to which the bacterial community within plant roots (root-endophytic compartment) influences plant-microbe interactions remains relatively unknown.” (Lines 36 – 38)

38 – I would have said “influences”

We have changed the word to “influences”. Please see above, we have changed this as you suggested. 

71 – suggest rewrite “relationships between the plant and the microbiome”

73 – long sentence – perhaps delete phrase after comma

To address both suggestions, we have rewritten this part as: 

“Second, few studies perturb the root-endophytic bacterial community with mechanisms which drastically impact the composition. To understand the relationships between the plant and the microbiome (34-37), experiments should include treatments which affect both the microbiome and the plant. Instead, many focus on perturbations that directly affect plant community dynamics.” (Lines 70 – 74)

78 – suggest rewrite “……….such as drought. This is important given that current predictions state that the intensity…………”

We have updated the sentence to read: 

“Many studies suggest that the root-endophytic bacterial community helps plants mitigate abiotic stress (32, 40-43) such as drought. This is important because the intensity and frequency of droughts are predicted to increase due to climate change (44, 45).” (Lines 76 – 79)

81 – suggest rewrite “………soil microbial diversity (as expected with antibiotic use) leads to……

81 – suggest rewrite “………..performance, and even fewer”

To address both suggestions, we have re-written this part as:

“Few studies have shown that a decrease in soil microbial diversity (as expected with antibiotic use) leads to decreases in plant performance, and even fewer have linked this to root-endophytic bacterial communities (2, 37).” (Lines 80 – 83)

95 – I might delete “must” (I know this is a part of feedback, but sentence structure seems awkward with it present)

We have removed the word “must”. The new sentence reads as:

“The term feedback within the scope of this study involves 2 steps: 1) the plant host perturbs the composition of the bacterial community, and 2) this differentiation affects the performance of the plant host (6).” (Lines 94 – 96)

98 – add comma after herbivore

We have included the comma: 

“Plant-soil feedbacks can predict co-existence of plant species since feedbacks are plant host-specific and can either be negative or positive (22) depending on the balance of negative effects of soil-borne pathogens, herbivores, and parasites compared to positive effects of beneficial soil microbes (60).” (Lines 96 – 99)

100 – add “plant” before the word “species”

We have taken this suggestion:

“…thus limiting dominance and competition among plant species.” (Lines 101)

109 – I haven’t read this book but 13% seems too high. Often one sees Samson and Knopf Bioscience 44:418-421 cited, with a reference to 1-4% prairies left. It is tricky – tables in this article show 1% prairie left in most states and more in a few states. These articles are also quite old – more prairie has been lost in the last 20 years. But alas, a really good reference doesn’t exist to my knowledge.

Yes, we haven’t been able to find a more recent citation. Most papers including this one published in 2017 still references the Samson and Knopf paper: North American tallgrass prairies represent one of the most widely destroyed and degraded ecosystems on Earth, with >90% of original prairie lost, and in some regions such as Illinois, >99% (Samson and Knopf, 1994; Samson and Knopf, 1996). (Barber et al 2017 Environmental Microbiology https://doi.org/10.1111/1462-2920.13785)

129 – need spaces between 1 and invasive

We have added the space. 

“We chose 5 prairie species, 4 natives: Monarda fistulosa (Wild Bergamot), Ratibida pinnata (Grey-head coneflower), Heliopsis helianthoides (Smooth oxeye), Conyza canadensis (Horseweed); 1 invasive which was listed as a noxious weed in Missouri (USDA 2019): Carduus nutans (Musk Thistle).” (Lines 127 – 130)

149 – suggest rewrite “(step 1 of the plant-feedback framework)”

150 - do you mean “sample closer to adult plants in field sites”??

Yes, we do mean in field sites. We have included that in the sentence to make it clearer. 

To address the last two comments we have re-written this sentence as:

“Step 1 of the plant-feedback framework is to create differentiated soil communities by either allowing plant hosts to grow in similar initial soil communities for a few months or sampling close to adult plants in field sites due to the short generation time and rapid community dynamics of microbial communities (54).” (Lines 149 – 152)

160 – Is this right? I would have expected soils to be pooled across experimental plots for those collected under the same plant species?

Yes, we did not pool the plots. We kept them separated and included that factor as a variable within our statistical model. 

201-2 - It is stated that plant performance was not affected when grown in field autoclaved or field soil, but I wasn’t sure whether this was a specific result of this study (if so, I’d expect a phrase like “See results” at the end) or whether this was a general statement (I’d expect to see a citation or at least xxx, unpublished data)

Yes, you are correct. We have added the reference to our figure in the supplements that demonstrate this result: 

“Antibiotics were chosen as a perturbation due to their ability to directly affect microbial communities by eliminating species from the communities without directly impacting plant growth (S1 Fig). Plant performance was not affected when grown in the presence or absence of antibiotics (S1B Fig).” (Lines 199 – 202)

219 Typo – “experiment” not “experimental”

Thank you for catching this typo. We have changed it to “experiment” instead of “experimented”. (Line 222)

229 – suggest rewrite “above-ground plant parts”

We have updated the sentence to reflect the suggested change:

“To calculate plant-soil feedback interaction coefficient (I_s), we used the dried biomass of both above-below ground plant parts and calculated the interaction using this equation (71):” (Lines 228 – 229)

248 – not sure of the meaning of root position

We did not standardize where we collected the root across all plants. We have re-worded the sentence to make this clear: 

“Since we wanted to limit the amount of cross contamination, we did not standardize the exact location of the extracted root sample across all plants.” (Lines 247 – 249)

250 insert “the” before “portion”

We have included the word “the” where suggested. (Line 258)

263 insert “the” before “microbial community”

We have included the word “the” where suggested. (Line 271)

306 perhaps insert “additionally” before “correlated” if the goal is to show this is an additional strength of the study

Yes, thank you for the suggested change. The new sentence is: 

“We additionally correlated the composition of the bacterial communities to plant performance to understand if differences in endophytic root bacterial compositions could explain differences in plant biomass.” (Lines 306 – 308)

308 insert “the” before “linear”

We have included the word “the” where suggested. (Line 318)

335 suggest rewrite “was better explained (as indicated by the r2 value) by plant identity…………..

We have changed the placement of the r2 value to the suggested placement. The new sentence is: 

“In a subset of samples which were non-autoclaved and non-antibiotic treated, the compositional differences in root bacterial community was better explained (as indicated by the higher r2 value) by plant identity (ADONIS p < 0.001, r2=0.11, Fig 2A) than by soil history (ADONIS p < 0.001, r2=0.03, Fig 2B).” (Lines 333 – 336)

357 there is a problem with the flow of this sentence

Thank you for bringing this to our attention. The new sentence is found below: 

“To determine if specific soil histories affected plant performance, we measured the dried biomass of all plants grown in each of the field-soil inocula.” (Lines 356 – 357)

394 replace comma with semi colon (or more generally think about how to reduce the complexity and structure of this sentence)

We reduced the complexity by creating two sentences. The new sentences are: 

“We can attribute plant performance to the soil biotic component because we controlled for potential variation in abiotic properties introduced by the small volume of soil history. Additionally, we assessed seedling growth in the same plant-inoculum combinations but with autoclaved inoculum.” (Lines 391 – 394)

407 – 410 I understand the meaning of the sentence but it is cumbersome to read.

To make this sentence easier to read, we removed some of the stats which can be found on the figure and referred to the figure just once. The new sentences are: 

“C. nutans (p < 0.013), H. helianthodies (p < 0.001), M. fistulosa (p < 0.01,), and R. pinnata (p < 0.0001), but not C. canadensis (p = 0.7), demonstrated a correlation between biomass and community similarity (Fig 4).” (Lines 406 – 408)

445-446 Reword. The reference to “we perturb” makes it sound like a methods sentence but clearly it is a results sentence

We have reworded this sentence: 

“To understand the specificity of the root endophytic microbial community to the host, we measured the change in root endophytic microbial community composition after exposure to antibiotics (ADONIS p = 0.26, r2 = 0.005) and the autoclave treatment (ADONIS p < 0.001, r2 = 0.07).” (Lines 444 – 447)

503 – It seems odd to emphasize the idea that individual plants within the same species are different before addressing the larger point that different host species structure the microbial community

Thank you for bringing this to our attention. We have rephrased to highlight the larger point and end with the individual plants. The new sentences are: 

“Within our study, we discovered that individuals within each plant host are more similar to each other than to different plant hosts (Fig 2A). Moreover, since we used natural variation within each plant host rather than artificially manipulating the plant genome and noticed large variation between individuals within each plant host indicating that root endophytic bacterial microbiome is not identical within all individuals of the same plant host (Fig 2A).” (Lines 499 – 504)

559 - should say “would become dominant.” Also, I think it is too much of a stretch to infer from a single lab study with autoclaving whether particular species would be more common after a fire. Many factors affect post-fire response.

Thank you for bringing this to our attention. We have changed the word “dominate” to “dominant”. Additionally, we have added the following sentences to highlight your point that more studies need to be conducted to understand the impact rather than using our study as the rule of thumb: 

“This result suggests that certain plant species would become dominant while others would become rare within the population after a prescribed fire.” (Lines 559 – 561)

“Additional studies are needed to understand the impact of current prairie restoration practices on root-endophytic bacterial communities and its impact on prairie community composition.” (Lines 564 – 566)

---

## [Decision Letter · Decision Letter 2]

29 May 2020

Prairie plants harbor distinct and beneficial root-endophytic bacterial communities

PONE-D-19-22834R2

Dear Dr. Adu-Oppong,

We are pleased to inform you that your manuscript has been judged scientifically suitable for publication and will be formally accepted for publication once it complies with all outstanding technical requirements.

With kind regards,

Cristina Armas

Academic Editor

PLOS ONE

Additional Editor Comments (optional):

Reviewers' comments:

Reviewer's Responses to Questions

**Comments to the Author**

1. If the authors have adequately addressed your comments raised in a previous round of review and you feel that this manuscript is now acceptable for publication, you may indicate that here to bypass the “Comments to the Author” section, enter your conflict of interest statement in the “Confidential to Editor” section, and submit your "Accept" recommendation.

Reviewer #2: All comments have been addressed

2. Is the manuscript technically sound, and do the data support the conclusions?

Reviewer #2: Yes

3. Has the statistical analysis been performed appropriately and rigorously? 

Reviewer #2: Yes

4. Have the authors made all data underlying the findings in their manuscript fully available?

Reviewer #2: Yes

5. Is the manuscript presented in an intelligible fashion and written in standard English?

Reviewer #2: Yes

6. Review Comments to the Author

Reviewer #2: (No Response)

7. PLOS authors have the option to publish the peer review history of their article (what does this mean?). If published, this will include your full peer review and any attached files.

Reviewer #2: No

---

## [Editor Report · Acceptance letter]

12 Jun 2020

PONE-D-19-22834R2 

Prairie plants harbor distinct and beneficial root-endophytic bacterial communities 

Dear Dr. Adu-Oppong:

I'm pleased to inform you that your manuscript has been deemed suitable for publication in PLOS ONE. Congratulations! Your manuscript is now with our production department. 

Kind regards, 

on behalf of

Dr. Cristina Armas 

Academic Editor

PLOS ONE